# Characterization of two affinity matured Anti-*Yersinia pestis* F1 human antibodies with medical countermeasure potential

Nileena Velappan[1], Sergei S. Biryukov[2], Nathaniel O. Rill[2], Christopher P. Klimko[2], Raysa Rosario-Acevedo[2], Jennifer L. Shoe[2], Melissa Hunter[2], Jennifer L. Dankmeyer[2], David P. Fetterer[3], Daniel Bedinger[4], Mary E. Phipps[5], Austin J. Watt[2], Rebecca J. Abergel[6,7], Armand Dichosa[1], Stosh A. Kozimor[8], Christopher K. Cote[2], Antonietta M. Lillo[1]*

1 Biosciences Division, Los Alamos National Laboratory, Los Alamos, NM, United States of America, 2 Bacteriology Division, United States Army Medical Research Institute of Infectious Diseases, Frederick, MD, United States of America, 3 Biostatisitics Division, United States Army Medical Research Institute of Infectious Diseases, Frederick, MD, United States of America, 4 Carterra, Walnut Creek, CA, United States of America, 5 Los Alamos National Laboratory, Center Alamos for Integrated Nanotechnologies, Los Alamos, NM, United States of America, 6 Chemical Sciences Division, Lawrence Berkeley National Laboratory, Berkeley, CA, United States of America, 7 Department of Nuclear Engineering, University of California, Berkeley, CA, United States of America, 8 Chemistry Division, Los Alamos National Laboratory, Los Alamos, NM, United States of America

* alillo@lanl.gov

**Data Availability Statement:** All relevant data are within the manuscript and its Supporting Information files. We have added the raw data for

## Abstract

*Yersinia pestis*, the causative agent of plague and a biological threat agent, presents an urgent need for novel medical countermeasures due to documented cases of naturally acquired antibiotic resistance and potential person-to-person spread during a pneumonic infection. Immunotherapy has been proposed as a way to circumvent current and future antibiotic resistance. Here, we describe the development and characterization of two affinity matured human antibodies (αF1Ig AM2 and αF1Ig AM8) that promote survival of mice after exposure to aerosolized *Y. pestis*. We share details of the error prone PCR and yeast display technology-based affinity maturation process that we used. The resultant matured antibodies have nanomolar affinity for *Y. pestis* F1 antigen, are produced in high yield, and are resilient to 37˚C stress for up to 6 months. Importantly, in vitro assays using a murine macrophage cell line demonstrated that αF1Ig AM2 and αF1Ig AM8 are opsonic. Even more importantly, in vivo studies using pneumonic plague mouse models showed that 100% of the mice receiving 500 μg of IgGs αF1Ig AM2 and αF1Ig AM8 survived lethal challenge with aerosolized *Y. pestis* CO92. Combined, these results provide evidence of the quality and robustness of αF1Ig AM2 and αF1Ig AM8 and support their development as potential medical countermeasures against plague.

Figs 2 through 5. The Materials and Method section is detailed enough to allow reproduction of each experiment described in the paper.

**Funding:** Most of this work was funded by the Los Alamos National Laboratory (LANL) Laboratory Directed Research and Development (LDRD) program, grant: LANL LDRD 20180005DR. Los Alamos National Laboratory, an affirmative action equal opportunity employer, is operated by Los Alamos National Security, LLC, for the National Nuclear Security Administration of the U.S. Department of Energy under contract DE-AC52-06NA25396. This work was also partly supported by JPEO-CBRND, through chemical biological defense funding.

**Competing interests:** The authors have declared that no competing interests exist.

**Abbreviations:** ACE2, Angiotensin converting enzyme; BSA, Bovine serum albumin; BSL, Biosafety level; CFU, Colony forming units; COVID-19, Coronavirus disease 2019; DLS, Dynamic light scattering; DMSO, Dimethylsulfoxide; EDTA, Ethylenediaminetetraacetic acid; ELISA, Enzyme-linked immunosorbent assay; epPCR, Error-prone polymerase chain reaction; F1, *Yersinia pestis* capsular protein fraction 1; Fab, Fragment antigen binding; IgG, Immunoglobulin G; $K_D$, Dissociation constant; MOI, multiplicity of infection; PE, Phycoerythrin; SARS CoV2, Severe acute respiratory syndrome coronavirus 2; scFv, Single-chain variable fraction; SPR, Surface plasmon resonance; TSB, Tryptic soy broth; VH, Variable heavy chain; VL, Variable light chain; WB, Wonder block; Y. pestis, *Yersinia Pestis*; YWB, Yeast wash buffer.

# Introduction

Antibody-based therapeutics, while popular in targeted treatment of cancer and autoimmune diseases [1–4], have only recently been deployed more broadly in the medical field [5–7]. It has been proposed that antibodies could be effective in fighting bacteria that are resistant to traditional antibiotics, provided that they bind the target with high specificity (only the pathogen is recognized) and high affinity (low dissociation constant [$K_D$], and low dosage requirement) [8–10], have favorable pharmacokinetics, and can be produced on an industrial scale. There are currently three FDA-approved antibodies (raxibacumab, obiltoxaximab, and bezlotoxumab) that protect by neutralization of bacterial exotoxins [11–13]. However, anti-bacterial antibodies could also target bacteria directly by promoting bactericidal activity, inhibition of biofilm formation/iron acquisition/adhesion, enhancing opsonophagocytosis, etc. [14]. For use in humans, it is preferable that therapeutic antibodies are human-derived or humanized to avoid an adverse immune response. Finally, a cocktail of antibodies is preferable to a single antibody to avoid mutation-based resistance. Remarkable recent examples of therapeutic antibody cocktails resulted from viral outbreaks. Monoclonal therapies were shown to be effective during recent Ebola virus outbreaks in Central and West Africa and clinical trials continue to test next-generation antibody strategies [15–17]. Anti-SARS-COV-2 antibodies were recently approved for emergency use by the FDA and played a significant role in the ameliorating the symptoms of COVID-19 [18]. When a single antibody is used, minimal mutation of the target antigen might result in failure of the antibody treatment [19,20]. However, massive mutations of a functional antigen (e.g., SARS-COV-2 spike proteins) are unlikely to happen, since they might result in loss of function (e.g., inhibition of interaction with ACE2 receptor and host cell invasion). Therefore, an antibody cocktail targeting multiple antigens or multiple regions (epitopes) of an antigen that plays an essential role in pathogenesis is better suited to fight resistance than a single antibody [21]. This oligoclonal antibody approach is analogous to the clinical administration of multiple classes of small-molecule antibiotics so as to attack resistance-prone bacteria simultaneously and reduce chances of emerging resistance [21,22].

*Yersinia pestis* is a gram-negative bacterium that causes various forms of plague [23,24]. The detection/surveillance of this bacterium and the diagnosis/treatment of plague remains relevant to both the public health and biodefense communities due to recent plague outbreaks [25–28], documented antibiotic resistance [29,30] and the potential for malicious use [31,32]. Fraction 1 (F1) is the dominant surface antigen of *Y. pestis*, produced during the initial intracellular phase of infection (at 34-37˚C) [33], and provides, together with other antiphagocytic factors, protection from host phagocytosis [34]. Antibodies directed against both F1V (F1 and V antigen) and F1 have shown efficacy in ameliorating the effects of F1-positive *Y. pestis* infection in animals [35–38]. However, with few exceptions [39,40], most of the potentially therapeutic and diagnostic antibodies described in existing literature are animal derived and/or polyclonal. We were the first group describing an orthogonal human antibody pair targeting F1, with potential for immuno- or radioimmunotherapeutic application as well as for point of care diagnostics [41,42]. These antibodies were selected from phage displayed human single-chain variable fragment (scFv) antibody libraries and the most promising ones were obtained and characterized as IgG1s.

*In vitro* selection of antibodies from display libraries, [43] (e.g., through phage [44,45]) and yeast [46] display technologies, enables identification of highly specific monoclonal antibodies, and offers advantages over more common methods (e.g., hybridoma technology) [47,48]. However, antibodies obtained by display technology usually require further optimization and characterization prior to their use as antibody-based medical countermeasures and diagnostics. Antibody affinity and stability maturation could be achieved by creating yeast-displayed

libraries of single monoclonal antibodies variants and screening them for clones with higher affinity and display efficiency than the parent antibodies [49]. Importantly, antibodies selected as single-chain variable fragment (scFv) or fragment antigen binding (Fab), need to be converted to full length immunoglobulins (IgG, the most common format for prophylactics and therapeutics), characterized for affinity and specificity, epitope mapped, and tested in vitro, ex vivo, and in vivo to assess suitability for use.

Here we describe maturation of two scFv antibodies previously selected in vitro for recognition of *Y. pestis* F1 antigen and whole bacterium. Matured scFvs [50] were converted to IgGs and characterized for affinity, stability, and epitope diversity. More importantly, we tested the ability of these antibodies to opsonize *Y. pestis* using an in vitro macrophage-based assay and to confer protection against *Y. pestis* infection using an in vivo mouse model of pneumonic plague. Our results support further development of these two antibodies for pre-exposure prophylaxes and possibly therapeutic use.

## Results and discussion

### Affinity maturation of anti-F1 antibody pair

A set of seven single-chain antibodies (scFv) against *Y. pestis* F1 antigen was previously described [51]. Among these antibodies, αF1sc 2 and 8 were chosen to be matured due to their ability to recognize distinct epitopes of the F1 antigen, their high affinity for F1, and their high production yield in IgG format [41,42]. Affinity maturation of these two antibodies was based on iterative sorting of yeast display libraries produced by error-prone PCR (epPCR) amplification αF1sc 2 and 8 clones (**Fig 1**). Mutant yeast display libraries of diversity ranging from 5 x $10^6$ to 8 x $10^6$ variants were obtained by homologous recombination of mutated antibody genes and linearized yeast display vector (Fig 1, steps 1 and 2). scFvs were expressed in tandem with SV5 tag and AGA2 yeast protein. These protein chimeras were displayed on the yeast surface in complex with surface protein AGA1 [46]. The antibody display level and antigen binding were determined by flow cytometry, upon staining with anti-SV5 IgG-phycoerythrin (PE) and purified biotinylated F1V dimer plus streptavidin-Alexa 635, respectively. Yeast sub-populations producing antibodies that displayed efficiently (high PE signal) and bound strongly to F1 antigen (high Alexa 635 signal), underwent multiple cycles of: a) sorting at progressively lower concentration of antigen; b) amplification; and c) staining (**Fig 1**, step 3). Monoclonal screening of final sorted population allowed us to single out the antibody mutant with the most efficient display (a proxy for foldability [42,52]) and affinity for the antigen (Fig 1, step 4). The gene encoding this antibody was purified from the displaying yeast, identified by sequencing (Fig 1, step 5) and subjected to the same treatment as the parental antibody (Fig 1, step 1 through 5).

An initial mutant library based on wild type αF1sc 2, plus three rounds of sorting, allowed us to select clone αF1sc 2 EP1-42, which includes the R234K mutation in the third complementarity-determining region (CDR3) of the variable heavy chain (VH). A second mutation library based on αF1sc 2 EP1-42, plus three rounds of sorting allowed the selection of clone αF1sc 2 EP2-3, which includes mutations R234K VH CDR3 and S185G VH CDR2. A third mutant library based on clone αF1sc 2 EP2-3, plus four rounds of sorting identified αF1sc 2 EP3-18 (αF1sc 2 AM2), which includes mutations Q31R VL CDR1, S35G VL CDR1, R234K VH CDR3 and S185G VH CDR2. The kinetic study of αF1sc 2 EP1-42 and αF1sc 2 EP2-3 suggests that these mutants do not have higher affinity for F1 than the original clone (similar $K_D$, **Figs 2 and S1**, S1 Table). It is likely that these mutants were selected because of their improved efficiency of display on yeast, as suggested by the increment of maximum antigen binding ($AB_{max}$). Although the data used for the kinetic studies were normalized for display efficiency

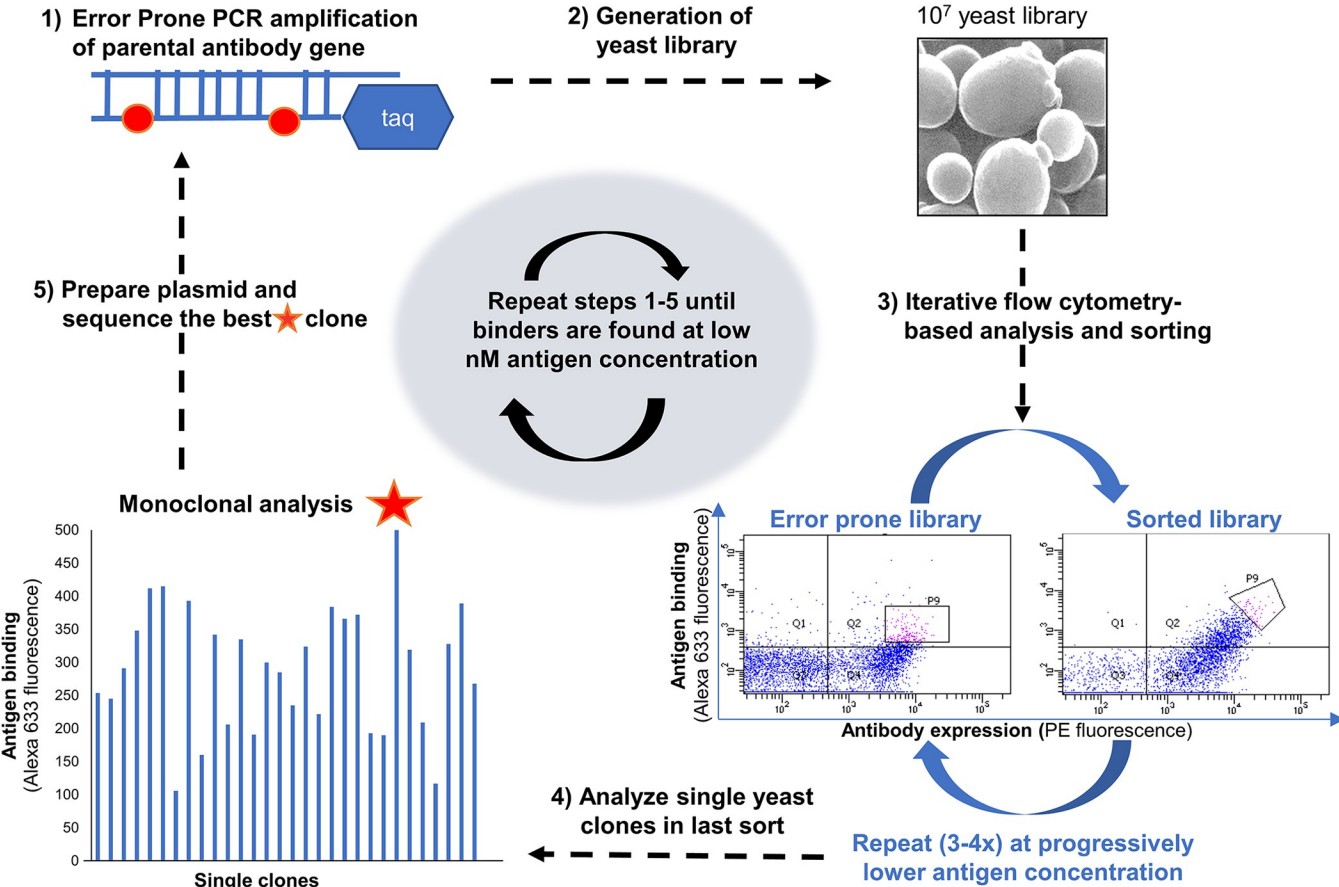

**Fig 1. Diagram of the affinity maturation process. 1)** Error-prone PCR introduces mutations in the original single chain antibody (scFv) gene. **2)** A library of ~ 1x10$^7$ diversity is created by co-transforming the error-prone PCR amplicon and the linearized display vector in yeast. **3)** The portion of the library with the highest level of yeast display (phycoerythrin -PE- fluorescence) and antigen binding (Alexa 633 fluorescence) is sorted by flow cytometry (2–5% of the entire population). Sorting is repeated 3 to 4 times at progressively decreasing antigen concentrations. **4)** Single clones in the last sorted population are analyzed one by one using sub-saturating amounts of antigen. **5)** The best binder (orange star) is identified by sequencing. Steps 1 through 5 are repeated until sorting at 1 nM antigen concentration is possible.

(aiming to obtain the same AB$_{max}$), it is possible that above a certain level of display the binding of anti-SV5 (a measure of display efficiency) reached saturation, thereby not allowing correct normalization of the binding data. These data suggest that the additional two mutations in the affinity maturation process result in a 3-fold higher affinity than the parent antibody.

An initial mutant library based on wild type αF1sc 8, plus three rounds of sorting, allowed us to select the best clone: αF1sc 8 EP1-19. Sequencing of this clone revealed mutation D33G VL CDR1. Three rounds of sorting of a second mutant library based on clone EP1-19 allowed us to select αF1sc 8 EP2-24 (αF1sc AM8), which includes mutations D33G VL CDR1 and H184R VH CDR2. Comparative kinetic studies of these two mutants versus αF1sc 8 (**Figs 2 and S1**) suggests higher affinity for F1V (lower $K_D$s) and higher display efficiency (higher AB$_{max}$). In conclusion, three mutant libraries and ten rounds of sorting resulted in identification of αF1sc AM2, whereas two mutant libraries and six rounds of sorting resulted in identification of αF1sc AM8. Final antibodies, together with respective parental antibodies (see sequences in **S1 Table**), were tested by flow cytometry for F1 binding at different of antigen concentrations (**Figs 2 and S1**). Yeast displayed αF1scAM2 and αF1sc AM8 appeared to have 3- and 9-fold higher affinity for F1 than the original clones respectively, showing that the

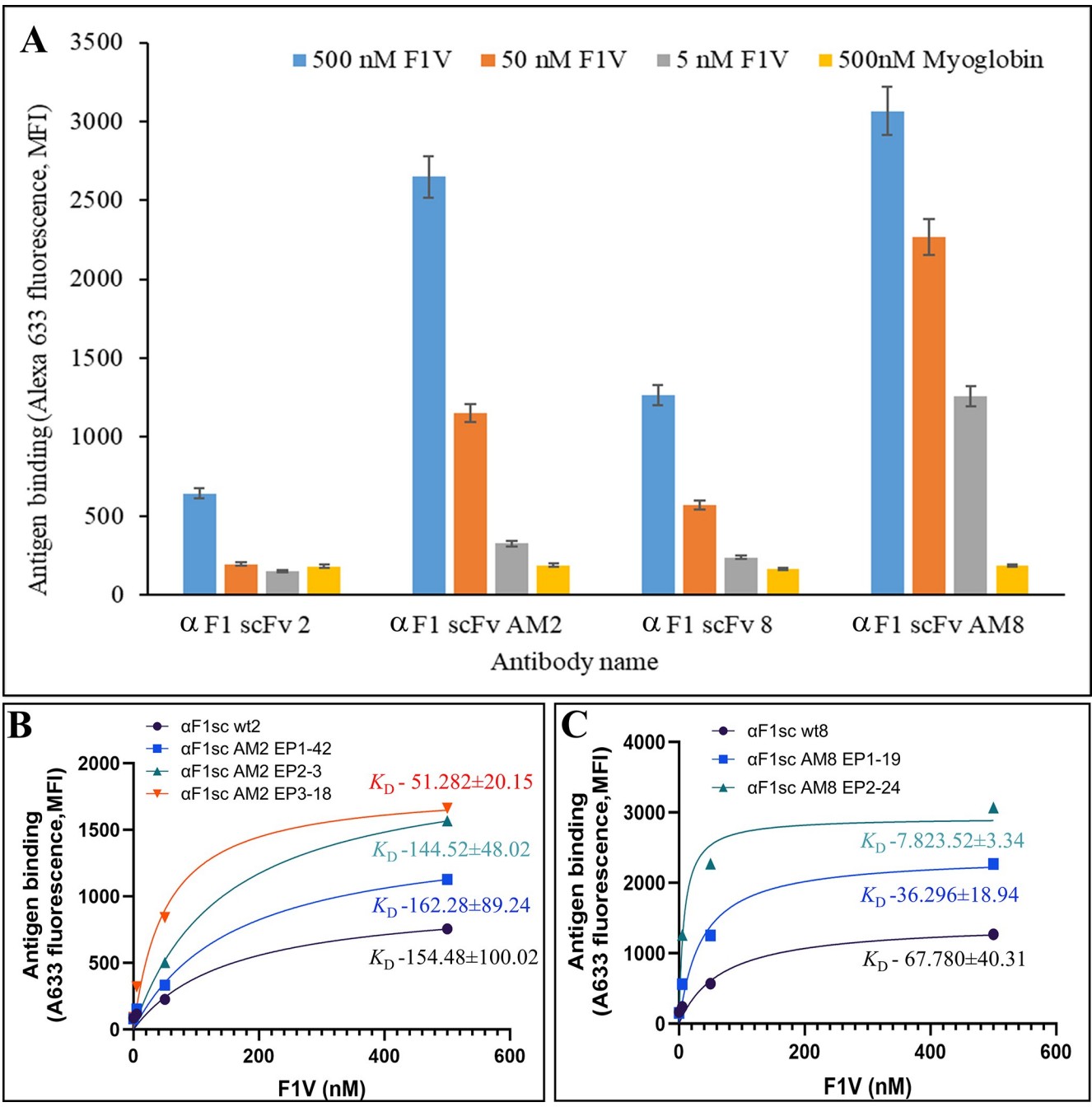

**Fig 2. Comparison of original and affinity matured yeast-displayed scFv antibodies.** (**A**) The yeast-associated fluorescence after incubation with biotinylated F1V and staining with streptavidin-APC (allophycocyanin) was measured by flow cytometry. Mean fluorescence intensity values at antigen concentrations of 500, 50, and 5 nM, and corresponding standard deviations (error bars) are reported for the original (αF1sc 2 and 8) and the affinity matured (αF1sc AM2 and AM8) antibody clones. 500 nM biotinylated myoglobin was used as a negative control antigen. (**B** and **C**) The same data reported in **A** were fit in the one-site binding equation [AB = ABmax*[Ag]/($K_D$+[Ag])] and $K_D$ resulting from the fit are indicated. The nomenclature EPX-Y in the legend denotes the error-prone library number (X) and the best clone number (Y) identified from this library. The best affinity matured αF1sc 2 clone (indicated for simplicity as αF1sc AM2 in **A**) was clone 18 from error-prone library 3 (αF1sc 2 EP3-18). The best affinity matured αF1sc 8 clone (indicated as αF1sc AM8 in **A**) was clone 24 from error-prone library 2 (αF1sc 8 EP2-24).

affinity maturation process was successful. Notice that $K_D$s obtained by flow cytometry, might be subject to error if the antigen concentration variation from the start to the end of yeast-antibody incubation period is not (as assumed) negligible. However, since the affinities of parent and affinity-matured antibodies were measured in the same way (i.e. they were subject to the same error), the magnitude of the difference in the affinity measured ($K_D$ of parental antibody/$K_D$ of matured antibodies) is reliable.

## Production and characterization of affinity matured anti-F1 IgGs

**IgG production.** Based on the sequences of αF1sc AM2 and αF1sc AM8, full-length antibodies (immunoglobulin G type 1, IgG1s) were produced by ATUM (www.atum.bio). Resulting antibodies were named αF1Ig AM2 and αF1Ig AM8 and their production yield was 0.26 mg/mL culture and 0.63 mg/mL culture, respectively (above the average, i.e. > 0.1 mg/mL culture).

**Antibody stability measurements by DLS.** Antibody drugs not only need to bind a target molecule, but they also need to be developable [53] i.e., they need to be expressed in high yield, have good solubility, be stable during long-term storage, and not be toxic. Although antibodies have evolved to be stable at 37˚C in serum, tracking antibody thermostability by dynamic light scattering (DLS) allows us to assess <u>antibody-specific</u> shelf life, propensity to aggregation, fragmentation, and potential generation of chemical by-products that could be toxic in vivo. Our DLS measurements were performed in the absence of preservatives (only saline solution) to assess the unaided resilience of our antibodies and shorten the duration of the experiment (stability at 37˚C) or avoid precipitation of adjuvants (high temperature stress). DLS has been used previously to demonstrate the structural integrity of monoclonal antibodies in solution [54]. DLS measurement of the hydrodynamic radius of both αF1Ig AM2 and αF1Ig AM8 in PBS before heat stress, revealed single peaks (100% of total species detected) corresponding to 12.12 and 14.91 nm radius, respectively. At 37˚C, hydrodynamic radius measured consistently in the 10–17 nm range for both antibodies, as well as the control antibody, over the course of a 150-day incubation period (**Fig 3**). Possible aggregates (higher hydrodynamic radius) appeared at the 60[th] day of incubation as peaks representing only 0.1% to 0.3% of the molecules detected. Since larger particles (aggregates) scatter light better than small particles, [55] these data suggest a negligible level of aggregation. We conclude that our anti-pestis antibodies could be stable beyond 150 days in harsh environments (37˚C is an extreme ambient temperature) when stored with preservatives.

IgGs were also subjected to DLS temperature scan from 37 to 55˚C. For αF1Ig AM2, temperatures and hydrodynamic radius sizes were as follows: 37˚C = 17.1 nm, 45˚C = 16.9 nm, and 55˚C = no signal. Since the signal from the antibody disappeared somewhere between 45˚C and 55˚C, we decided to take more measurements between 45˚C and 55˚C to more accurately ascertain at what temperature the aggregation occurred. Unfortunately, due to constrains in antibody availability we were only able to perform a more accurate assay with αF1Ig AM8. For this antibody the temperatures and sizes measured were: 37˚C = 15.3 nm, 45˚C = 15.7 nm, 47˚C = 14.8 nm, 49˚C = 14.6 nm, 51˚C = 16.6 nm, 53˚C = 6.7 nm, and 55˚C = no signal. Lack of signal at 55˚C suggests antibody precipitation out of solution at Tm = 55 for both antibodies. We concluded that at least αF1Ig AM8 is thermostable up to 51˚C. We were able to assess that aggregation and fragmentation at 0.3 uM antibody concentrations is negligible up to 45˚C (αF1Ig 2) and 51˚C (αF1Ig AM8). These thermostability data together with the high production yields (section above) suggest good manufacturability.

**Kinetics study.** A kinetic study of affinity matured IgGs was performed by ELISA and SPR. Affinity constants ($K_D$s), together with affinity improvements, are reported in **Table 1**.

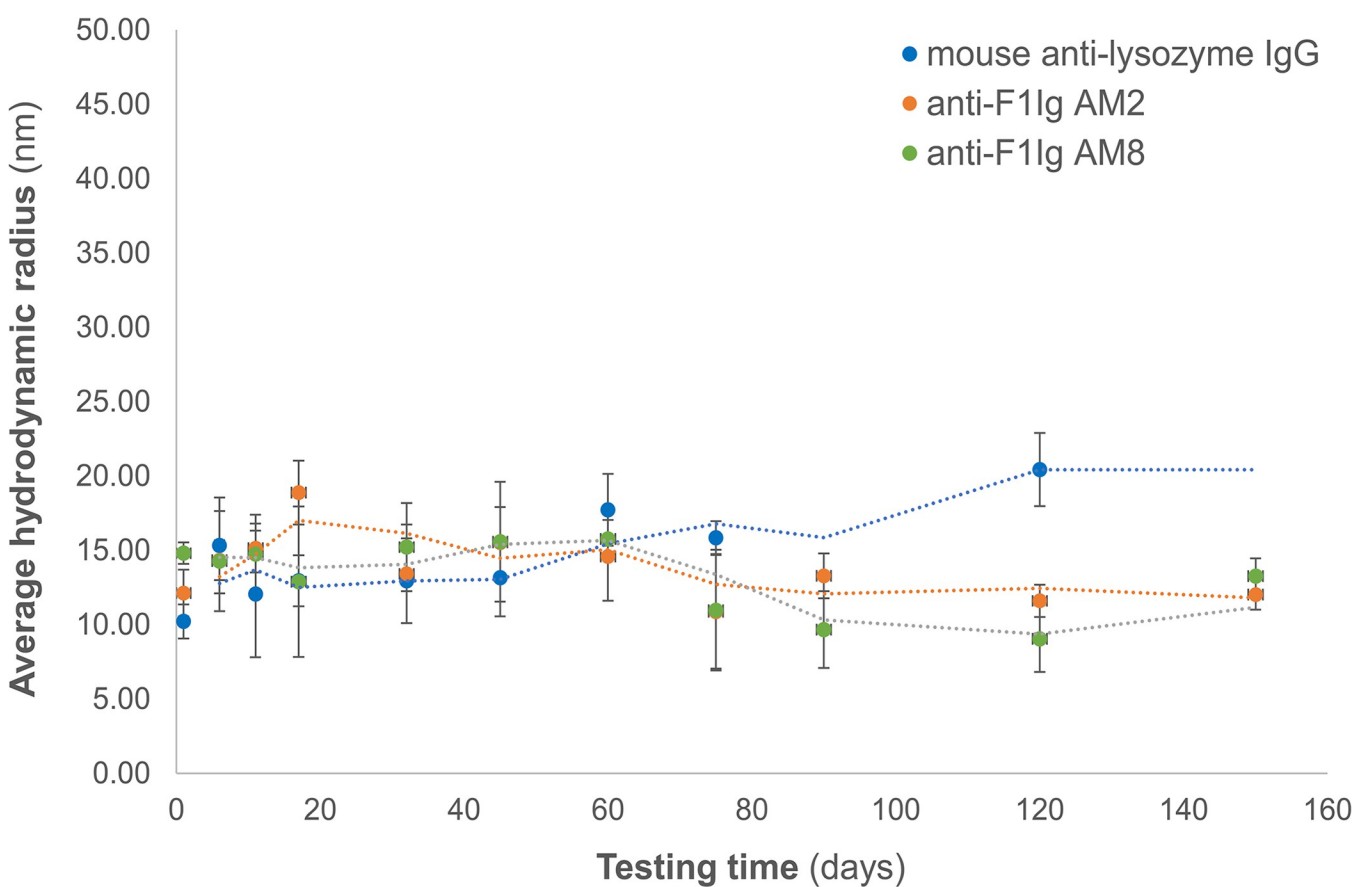

**Fig 3. Antibody stability by dynamic light scattering analysis.** The hydrodynamic radius of anti-F1 affinity matured antibody 2 and 8 (αF1Ig AM2, orange; αF1Ig AM8, green) was ~14 nm and did not change after 150 days storage at 37˚C in PBS. These results indicate lack of protein aggregation in the conditions tested. A mouse anti-lysozyme IgG stored at 4˚C for the duration of the experiment was used as a control (blue). In this figure, two per moving average trendline is indicated as dotted lines.

Flow cytometry-based kinetics of yeast displayed scFvs are also reported here for ease of comparison (better described in Figs 2 and S1).

Capsular antigen F1 is expressed on *Y. pestis* surface at >34˚C; therefore, for ELISA-based kinetic studies of αF1 AM2 and 8 we used *Y. pestis* A1122 grown at 37˚C and immobilized on plastic to generate a lawn of cell-associated F1 antigen and perform whole cell ELISA (S2A Fig, top). Furthermore, in order to execute the experiments under Biosafety Level 1 conditions, *Y. pestis* cells were fixed (i.e. killed) knowing that fixation does not affect the F1 interaction with our antibodies [41]. Functional affinity for cell-expressed F1 was measured at various concentrations of antibody and data were fit in the one-site specific binding equation (Eq #1 in *Materials and Methods*, Fig 4A). αF1Ig AM2 and αF1Ig AM8 appear to have 2- and 3-fold higher affinity for capsule-embedded F1 than the parental clones (Table 1). The difference in affinity increments between yeast displayed scFvs obtained by flow cytometry (3 and 9-fold affinity increment for αF1sc 2 and αF1sc 8 respectively) and soluble IgGs measured by ELISA, is likely due to different antibody formats (more below) and antigen formats. Specifically, soluble recombinant F1V antigen was used in the flow cytometry assays, and native F1V was used in ELISA. The recombinant F1V used for flow cytometry sorting is dimer-enriched antigen purified from *E. coli*. In this construct the signal peptide on the native F1 gene is removed and a dipeptide linker (EF) is introduced at junction between F1 and the V proteins. The resulting

**Table 1. Affinities of anti-F1 antibodies for F1 antigen.**

| Antibody[a] | Antigen format /assay method | Affinity ($K_D$, nM) | Affinity improvement[b] |
|---|---|---|---|
| **αF1sc 2** | SBV[c]/FC[d] | 154.5 ± 109.4 | - |
| **αF1sc AM2** | SBV/FC | 51.3 ± 20.2 | 3-fold (based on flow cytometry) |
| **αF1Ig 2** | CD[e] /ELISA[f] | 0.4 ± 0.0 | - |
| | S[g] /SPR | 1.9 ± 0.8 | |
| **αF1IgAM2** | CD/ELISA | 0.2 ± 0.0 | 2-fold (based on ELISA) |
| | S/SPR | 2.7 ± 1.3 | None (based on SPR) |
| **αF1sc 8** | SBV/FC | 67.7 ± 40.4 | - |
| **αF1sc AM8** | SBV/FC | 7.8 ± 3.4 | 9-fold (based on flow cytometry) |
| **αF1Ig 8** | CD/ELISA | 0.3 ± 0.0 | - |
| | S/SPR | 7.4 ± 2.5 | |
| **αF1Ig AM8** | CD/ELISA | 0.1 ± 0.0 | 3-fold (based on ELISA) |
| | S/SPR | 1.9 ± 0.7 | 4-fold (based on SPR) |

[a] Either αF1sc (anti-F1 single chain antibody underline{expressed on yeast}), or αF1Ig (anti-F1 immunoglobulin G type 1, IgG1, underline{in solution}). The number indicates the clone.
AM = affinity matured.

[b] Parental IgG $K_D$/affinity matured IgG $K_D$.

[c] Soluble and biotinylated F1 in complex with V antigen (F1V dimer enriched).

[d] Flow cytometry analysis of yeast-displayed scFv.

[e] CD = cell (*Y. pestis*)-displayed F1.

[f] ELISA = Enzyme-linked immunosorbent assay.

[g] Soluble F1 antigen (monomer-enriched).

chimera is produced as soluble protein. Although this recombinant protein was designed to mimic the native protein, the required manipulations could have potentially altered the structure and stability of the proteins in subtle ways. During affinity maturation, we used the

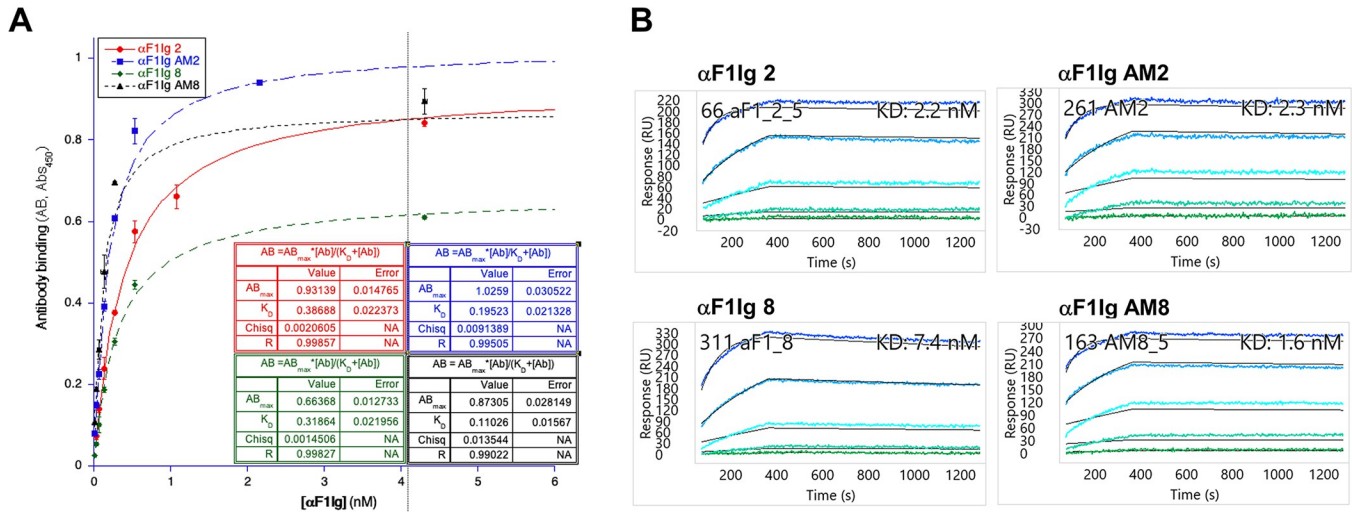

**Fig 4. Kinetic study of purified anti-F1 IgGs by ELISA and SPR.** (**A**) ELISA data obtained for original anti-F1 IgGs (αF1Ig 2 and 8) or affinity matured (αF1Ig AM2 and 8) antibodies are shown. Experiments were performed in triplicate and data points averaged. The standard deviation of each data point was calculated and shown as the error bar. Data were fitted to one-site specific binding equation. AB = antibody-F1 binding; $AB_{max}$ = antibody-F1 binding at saturation; $K_D$ = dissociation constant = half saturating antibody concentration ([Ab]). (**B**) Representative surface plasmon resonance sensograms for each parental and affinity matured antibody are shown.

recombinant F1V fusion protein. It can be hypothesized that the increase in affinity might be due to additional contact points and/or structural adjustments occurring in the yeast displayed scFv to the presence of the V antigen in the chimera, contributed to the more pronounced changes in affinity. In vitro antibody selection and maturation technologies are highly sensitive to antigen formats, hence differences between recombinant fusion protein and native cell surface protein binding should be anticipated. Nevertheless, we can state that αF1Ig AM2 and αF1Ig AM8 both recognize native and recombinant F1 and that, based on ELISA, the affinity gain achieved by the scFv display-based maturation method seems to be preserved when the antibody format transitions from single chain to full length IgG.

Affinity characterization was also performed by surface plasmon resonance (SPR, **Figs 4B** and **S2C**). For this analysis, αF1Ig AM2, αF1Ig AM8 and the corresponding parental IgGs were immobilized on sensor chips at various concentrations and exposed to various concentrations of <u>monomer-enriched recombinant F1</u> antigen (**S2A Fig,** bottom). αF1Ig AM8 was found to have 4-fold higher affinity than parental αF1Ig 8, however no affinity increment was observed for αF1Ig AM2 *vs* αF1Ig 2. A major cause of this discrepancy could again be the difference in the antigen format used in the two assays. During flow-based kinetic studies (and the affinity maturation process) we used dimer-enriched F1V complex, whereas for SPR kinetics we used monomer-enriched F1. It is possible that αF1 antibody 2 binds at the intersection between F1 and V antigens and that the antibody changes during affinity maturation were tailored to the presence of V antigen. The lack of V antigen in the antigen used for the SPR assay (monomeric F1, <u>not</u> F1V) might explain why this assay failed to reveal affinity maturation. Also notice, that although SPR is the gold standard in affinity measurements, the ELISA used here reflects a more real-world scenario (i.e. antibody interacting with *Y. pestis*-displayed F1 antigen) than the SPR assay. Therefore, ELISA data should weigh more in the decision of whether αF1Ig AM2 has indeed been matured.

Based on these observations we could conclude that both matured IgGs have higher affinity for F1 antigen than parental IgGs. We do recognize that the increment in affinity was not as high as expected, however please note that increased manufacturability is an additional goal of the maturation process. This is the reason why during maturation we sort yeast that are not only strong antigen binders but also strong displayers. High level of display on the yeast surface is generally a proxy for high stability (e.g. thermostability), foldability and low aggregation (all associated to manufacturability). Therefore, matured antibodies are likely to have higher stability than the parental antibodies. Although in this work the stability of matured antibodies was not measured side by side with the parental counterpart, we previously reported that αF1Ig 2 and 8 start to degrade after 2.5 weeks storage at 37˚C, whereas matured αF1Ig AM2 and AM8 (this paper) show no sign of aggregation after 150 days.

In summary, we can safely conclude that our yeast display-based affinity maturation process was successful and resulted in improved variants of parental antibodies αF1Ig 2 and αF1Ig 8, i.e. αF1Ig AM2 and aF1Ig AM8.

## Potential use of αF1Ig AM2 and αF1Ig AM8 as prophylactic treatments

**αF1Ig AM2 and αF1Ig AM8 anti-F1 antibodies are opsonic in vitro.** Opsonization is antibody-based phagocytosis of bacteria by phagocytes. Prior to examining the mAbs for pre-exposure prophylaxis, we evaluated the effect of the αF1Ig AM2 and αF1Ig AM8 on the interaction of *Y. pestis* with RAW264.7 cells (murine macrophage-like cells). In these assays, both antibodies were significantly opsonic (**Fig 5**). The bacteria pretreated with either mAb were phagocytosed to a significantly greater extent than the negative control mAb (α-influenza αM2 IgG Z3). The positive control mouse-derived αF1 mAb F1-04-A-G1 appeared to be more

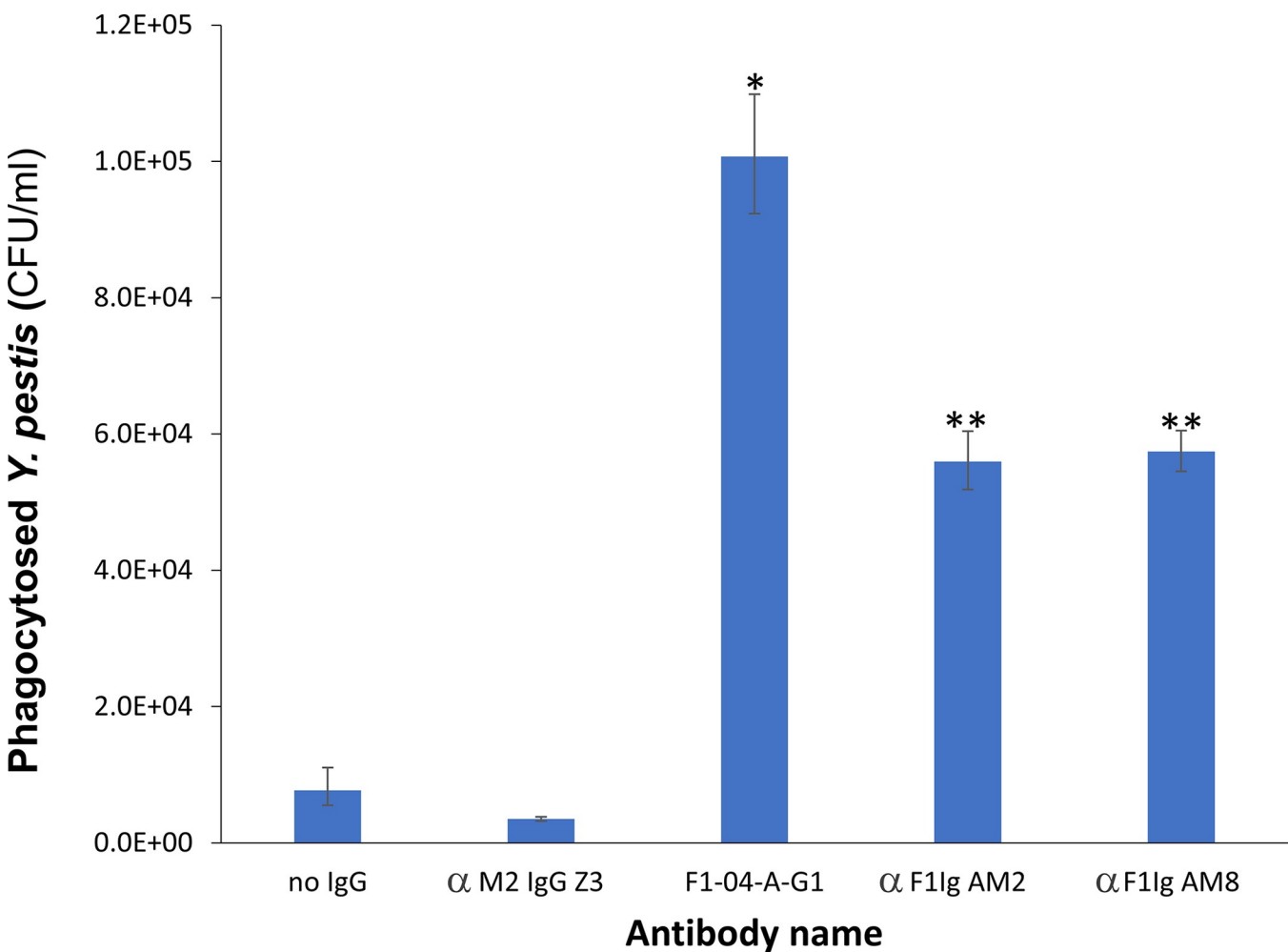

**Fig 5. The anti-F1 monoclonal antibodies are opsonic in nature.** As demonstrated in an *in vitro* gentamicin protection assay the mouse derived α-F1 mAb (F1-04-A-G1) and the human derived αF1sc AM2 and αF1sc AM8 mAbs were opsonic. RAW264.7 cells were infected with *Y. pestis* CO92 *pgm*- pPst- at an MOI of approximately 10 CFU. (*) indicates P <0.0001 when compared to *Y. pestis* (no antibody) alone and P < 0.0001 when compared to anti-influenza M2 (αM2 IgG Z3) negative control, (**) indicates P = 0.0007 when compared to *Y. pestis* alone and P < 0.0001 when compared to anti-influenza M2 negative control. This is the average of 2 experiments and representative of 5 experiments total.

potently opsonic than αF1Ig AM2 and αF1Ig AM8, however this might be an artifact due to a better response of the mouse-derived RAW264.7 cells to a murine antibody (**Fig 5**).

**αF1Ig AM2 and αF1Ig AM8 anti-F1 antibodies protect mice from pneumonic plague.** Initially we tested the antibodies at 500 µg doses. The mAbs were administered intraperitoneally approximately 18 h prior to exposure to aerosolized *Y. pestis* CO92. In this first experiment, all control mice (PBS-treated, normal mouse IgG-treated, or α-influenza M2-treated) had succumbed to disease or were euthanized by day 4 post-exposure to aerosolized bacteria. However, 100% of the mice receiving mouse-derived αF1 mAb F1-04-A-G1, αF1Ig AM2 or αF1Ig AM8 survived the infection with no clinical signs of disease (**Table 2, experiment #1**). We also performed several dose-down studies and demonstrated that partial protection was achieved with either a 400 µg dose or 187.5 µg dose of αF1Ig AM2 (**Table 2, experiments #2 and #3**). However, doses of 125 µg or lower offered no overall protection from pneumonic plague (**Table 2, experiment #4**), mirroring the results obtained when mice were pre-treated with the mouse-derived αF1 mAb F1-04-A-G1. These dose-down studies did not include

**Table 2. *In vivo* experiments using the BALB/c mouse model of pneumonic plague to demonstrate protection of antibody therapy when given as pre-exposure prophylaxis.**

| Expt # | Antibody | Antibody amount[1] | Challenge dose | LD50[2] | # of mice challenged | # of mice survived | Survival P value[3] | TTD Median (Q1, Q3)[4] | TTD P value[3] |
|---|---|---|---|---|---|---|---|---|---|
| 1 | αF1Ig AM2 | 500 µg | $5.32 \times 10^5$ | 7.8 | 10 | 10 | <0.0001 | >21 | <0.0001 |
| | αF1Ig AM8 | 500 µg | $5.32 \times 10^5$ | 7.8 | 10 | 10 | <0.0001 | >21 | <0.0001 |
| | αM2 IgG Z3 | 500 µg | $5.32 \times 10^5$ | 7.8 | 10 | 0 | 1 | 3.0 (3.0,3.0) | 0.32 |
| | F1-04-A-G1[5] | 500 µg | $5.32 \times 10^5$ | 7.8 | 10 | 10 | <0.0001 | >21 | <0.0001 |
| | normal mouse IgG | 500 µg | $5.32 \times 10^5$ | 7.8 | 5 | 0 | 1 | 3.0 (3.0,3.0) | 0.56 |
| | PBS | n/a | $5.32 \times 10^5$ | 7.8 | 5 | 0 | 1 | 3.0 (3.0,3.0) | 1 |
| 2 | αF1Ig AM2 | 400 µg | $2.95 \times 10^5$ | 4.3 | 8 | 6 | 0.007 | 4.0 [6] | 0.0008 |
| | F1-04-A-G1 | 400 µg | $2.95 \times 10^5$ | 4.3 | 8 | 8 | 0.0002 | >21 | <0.0001 |
| | PBS | n/a | $2.95 \times 10^5$ | 4.3 | 8 | 0 | 1 | 3.0 (3.0,4.0) | 1 |
| 3 | αF1Ig AM2 | 187.5 µg | $4.75 \times 10^5$ | 7.0 | 8 | 6 | 0.007 | 12.5 [6] | <0.0001 |
| | F1-04-A-G1 | 187.5 µg | $4.75 \times 10^5$ | 7.0 | 8 | 8 | 0.0002 | >21 | <0.0001 |
| | PBS | n/a | $4.75 \times 10^5$ | 7.0 | 8 | 0 | 1 | 4.0 (3.5,4.0) | 1 |
| 4 | αF1Ig AM2 | 125 µg | $1.4 \times 10^6$ | 20.6 | 8 | 0 | 1 | 4.0 (4.0,4.0) | 0.0008 |
| | F1-04-A-G1 | 125 µg | $1.4 \times 10^6$ | 20.6 | 8 | 0 | 1 | 5.0 (4.5,5.0) | 0.0002 |
| | αF1Ig AM2 | 50 µg | $1.4 \times 10^6$ | 20.6 | 8 | 0 | 1 | 4.0 (4.0,4.0) | 0.0006 |
| | F1-04-A-G1 | 50 µg | $1.4 \times 10^6$ | 20.6 | 8 | 0 | 1 | 4.0 (4.0,4.0) | 0.0006 |
| | αF1Ig AM2 | 25 µg | $1.4 \times 10^6$ | 20.6 | 8 | 0 | 1 | 4.0 (4.0,4.0) | 0.0037 |
| | F1-04-A-G1 | 25 µg | $1.4 \times 10^6$ | 20.6 | 8 | 0 | 1 | 4.0 (4.0,4.0) | 0.0008 |
| | PBS | n/a | $1.4 \times 10^6$ | 20.6 | 8 | 0 | 1 | 3.0 (3.0,3.0) | 1 |

[1]Antibodies or PBS were given as a single intraperitoneal bolus approximately 18 h prior to exposure to aerosolized *Y. pestis* CO92.

[2] Based on an estimated $LD_{50}$ of $6.8 \times 10^4$ inhaled CFU.

[3] Statistical analyses compared to mice receiving PBS.

[4] TTD represents time to death or euthanasia; values in parentheses represent the 25 and 75 percentile.

TTD for each group.

[5] F1-04-A-G1 is a mouse derived anti-F1 mAb used as a positive control.

[6] This value represents the mean TTD or euthanasia of animals that died. The Median TTD (Q1, Q3) could not be computed based on the number of animals succumbing to infection.

αF1Ig AM8 since data from a previous set of preliminary experiments (unpublished data, available upon request) showed that this antibody performs slightly worse than αF1Ig AM2. While the effective doses determined in these experiments are clearly higher than desirable for human use, the BALB/c mouse model of pneumonic plague is quite stringent, and these protective data serve as proof-of-concept for further optimization and/or antibody engineering that may be required for optimal drug product. Additionally, these anti-F1 mAbs were evaluated as single medical countermeasures, but there has been documented synergy when anti-F1 and anti-LcrV mAbs are used in combination. Hill et al. [36] demonstrated that greater protection from bubonic plague was associated with an anti-F1 mAb when combined with antibodies directed against the LcrV protein, a multi-functional protein that is a component of the type three secretion injectisome and essential for virulence [56,57]. It is also conceivable that simultaneous administration of αF1Ig AM2 and αF1Ig AM8 might result in a synergistic effect due to orthogonal binding of these two antibodies to the F1 antigen [41], resulting in lowering of the protective dosage of the antibody cocktail. Finally, it is important to note that due to inherent variability associated with the aerosolization of bacteria, the calculated inhaled dose was approximately 3-fold higher in the experiment evaluating the lowest dose ranges of antibodies

(**experiment #3 & #4**). These lower doses of administered antibodies did extend the time to death or euthanasia compared to PBS ($p < 0.0037$), but the biological relevance of this delay remains equivocal. Thus, under the right conditions, lower concentrations of anti-F1 mAbs may be even more effective than we observed. Our efforts to develop novel anti-F1 mAbs will be beneficial to both the biodefense and public health research communities. While the overwhelming number of human plague cases have been caused by encapsulated strains of *Y. pestis*, nonencapsulated strains have been isolated in nature [58,59]. However, very few documented cases of human disease initiated by these strains exist [60]. Nevertheless, nonencapsulated strains remain virulent in animal models of plague and accordingly, other antigenic targets (e.g. LcrV) should always be considered when developing a medical countermeasure against plague [61–66]. We envision an effective anti-F1 mAb to be beneficial in most cases of plague, but a cocktail approach would certainly provide the most complete protection against a number of different *Y. pestis* strains with differing antigen profiles (e.g. nonencapsulated).

## Conclusion

Here, we describe development of two antibodies (αF1Ig AM2 and αF1Ig AM8) that show potential as medical countermeasures against pneumonic plague. Because these two antibodies bind orthologous epitopes on the F1 protein of *Y. pestis* and each promoted antibody-mediated opsonization, together they could be candidates for developing a passive immunity cocktail for post-exposure or prophylactic use. Our in vivo studies demonstrated a significant protective effect when delivered prior to exposure. This protection in the mouse mode of pneumonic plague appears to be dose-dependent. However, due to the inherent variability associated with small-particle aerosol delivery of *Y. pestis* resulting in different doses of delivered bacteria, dose-dependence is not strongly supported. Further, these antibodies have favorable manufacturability characteristics: they can be produced in large amounts and have a long shelf-life.

In addition to demonstrating their potential value to national security, we also present the methodology to develop such high-quality reagents. Yeast display–based affinity maturation of scFv antibodies, production of full length IgGs, DLS measurements to analyze antibody stability, in vitro characterization to understand mode of action, and mouse model-based assessment of efficacy have proven to be a valuable antibody development/characterization pipeline.

While the results presented here may provide a solid pharmaceutical foundation, future work should be focused on elucidation of pharmacokinetics and pharmacodynamics, such as testing of different treatment strategies (e.g., dosing, dosing schedules, combination, or layered defense strategies) and characterization of human tissue cross-reactivity. Such elucidative experiments could further evaluate the potential of these two antibodies in fighting natural or manmade plague outbreaks.

## Materials and methods

### Yeast display-based affinity maturation of antibodies αF1sc 2 and αF1sc 8

**Generation of mutant libraries.** Mutant libraries of αF1sc 2 and 8 were created and sorted for selection of higher affinity/expression antibodies variants following a previously published protocol [49]. Briefly, the antibody genes were amplified from previously described yeast display plasmids pDNL6-αF1sc 2 and pDNL6-αF1sc 8 [41] by error-prone polymerase chain reaction (epPCR) using PNL6 forward and reverse primers (`GTACGAGCTAAAAGTA CAGTG` and `TAGATACCCATACGACGTTC`). The epPCR reaction mixture included a 5-fold excess dATP/dGTP versus dTTP/dCTP (2 mM and 0.4 mM, respectively), DMSO (2%), MnCl$_2$ (0.2 mM), MgSO$_4$ (0.2 mM) and Taq polymerase (NEB, cat# M0273L). The

thermocycling PCR program was the following: initial denaturation (95°C, 2 min), 30 cycles of denaturation (95°C, 30 sec), annealing (58°C, 30 sec), polymerization (68°C, 60 sec), and final extension (68°C, 7 min). Four hundred to 500 μL epPCR reaction generated 10–20 μg of insert DNA for both antibodies. Plasmid pDNL6-CBH1 [49,67] was digested with restriction enzymes *BssH*II, *Nhe*I, and *Nco*I. The quality of the plasmid and digestion was verified by gel electrophoresis. Of note, separation of plasmid and CBH1 insert was not necessary since formation of a library of pDNL6-antibody mutant plasmids is not based on ligation but on recombination upon yeast transformation. Both the epPCR inserts and the cut vector were purified (Qiagen, #28104) prior to yeast transformation. Preparation of $10^7$ mutant library used 20 μg of vector DNA and 10 μg of insert DNA. The yeast transformation protocol used 100 mL of *S. cerevisiae* EBY$_{100}$ cells at > OD$_{600}$ 2.0 (4 x $10^7$ cells/mL). The lithium acetate-based chemical transformation protocol was performed as described before [68].

**Yeast library selection.** Transformed yeast were grown in selective medium (SD/CAA) for two days at 30°C and yeast induction was performed in SGR/CAA medium at 20°C for 16–20 h. The SD/CAA medium was composed of 5 g/L casamino acids (-ade, -urs, -trp), 20 g/L dextrose, 1.7 g/L yeast nitrogen base, 5.3 g/L ammonium sulfate, 10.19 g/L Na$_2$HPO$_4$.7H$_2$O, and 8.58 g/L NaH$_2$PO$_4$. The SGR/CAA medium was the same as selective medium except for dextrose, which was substituted with 20 g/L galactose, 20 g/L raffinose, and 1 g/L dextrose. Five hundred microliters of SD/CAA overnight culture was added to 5 mL of SGR/CAA for induction of antibody display. After overnight induction, 200–1000 μL of the yeast was washed with yeast wash buffer (YWB- 1X PBS, 0.5% BSA) by centrifugation. The yeast cells were then incubated with F1V dimer (BEI Resources, NR-2563) which was biotinylated using NHS-LC-LC biotinylation kit (ThermoFisher #21343). Incubation with biotinylated antigen was allowed to proceed for 30–60 min with rotation on a bench top rotator. Yeast cells were washed twice and stained with phycoerythrin (PE) conjugated anti-SV5 IgG (ThermoFisher #37–7500, 1 μg/mL, to assess scFv display levels) and streptavidin-Alexa 635 (ThermoFisher #S21375, 5μg/mL, to detect binding to biotinylated antigen). Flow cytometry-based analysis of yeast staining and subsequent sorting of best F1-binding/scFv-expressing cells were performed using BD FACS Aria. The initial mutant libraries were incubated with 500 nM bio-F1V, analyzed, and sorted. After three rounds of sorting, yeast populations binding F1V at 50 nM were sorted. Single clones from the third sort were subjected to monoclonal analysis to identify best clones. Plasmid DNA from these clones was prepared and another set of epPCR was performed to introduce further mutations. Two sets of mutant libraries and six rounds of sorting was performed to identify the αF1sc 8 variants that binds the F1V antigen with 1 nM affinity. Similarly, three sets of epPCR and ten rounds of sorting were performed to identify the best αF1sc 2 variants that also recognized F1V antigen at 1 nM. Single clones from the two best variant populations were analyzed using 5 nM F1V antigen. Plasmid DNA from each of the best monoclonals was prepared and sequenced by Sanger sequencing.

**Kinetic characterization of yeast displayed affinity matured antibodies.** Affinity measurements were conducted using yeast displayed affinity-matured or parent antibodies incubated with serial dilutions of biotinylated-FIV (same antigen format used in affinity maturation process). Yeast cells were analyzed for expression and antigen binding by flow cytometry. Antigen-antibody binding was reported by the Alexa 633 fluorescence (medium fluorescence intensity, MFI; 635 excitation, 660/20 emission). This yeast-associated fluorescence derives from streptavidin-Alexa 633 conjugate, interacting with the yeast-bound biotinylated antigen. Antigen-antibody binding (Alexa 633 fluorescence MFI) at each concentration of antigen was normalized to the level of antibody display on yeast reported by Phycoerythrin fluorescence (MFI, 561 excitation, 582/15 emission). The resulting data (antibody-antigen binding and corresponding antigen concentration) were fit to the one-site specific binding

equation using GraphPad software.

$$AB = AB_{max}*[Ag]/(K_D + [Ag]) \qquad \text{Eq1}$$

In this equation AB is antigen-antibody binding (Alexa 635 MFI, variable), $AB_{max}$ is the antigen-antibody binding (Alexa 635 MFI) at antibody-binding saturation (constant), [Ag] is the initial antigen concentration (variable), and $K_D$ is the apparent antibody-antigen dissociation constant.

## Production of affinity matured anti-F1 IgGs and their characterization

**IgG preparation.** Wild-type and affinity-matured scFvs were converted to IgGs by inserting the amino acid sequences corresponding to the variable heavy (VH) and variable light (VL) antibody regions into a standard IgG1 scaffold. The resulting protein sequences were submitted to ATUM (Newark, CA, USA) for codon-optimized back-translation, gene synthesis, and expression as full-length IgG1 antibodies in HEK293 cells. IgG were received frozen from ATUM and stored in small aliquots at -80˚C before use in various assays.

**Stability study.** αF1Ig AM2 and αF1Ig AM8 were separately incubated in PBS pH 7.4, in a total volume of 1 mL, at concentrations of 0.87 mg/mL and 0.97 mg/mL, respectively. Antibody solutions were stored in Eppendorf tubes inside a humid chamber at 37˚C and periodically analyzed for stability by dynamic light scattering (DLS). DLS measurements or the hydrodynamic radius of each antibody and a control antibody (mouse anti-lysozyme IgG stored at 4˚C in PBS pH 7.4) were taken immediately after obtainment of antibody solutions (day 0) and then over the course of 150 days' storage at 37˚C. Measurements were performed using a Malvern Zetasizer Nano ZS with the following settings: material = protein, refractive index = 1.45, dispersant = water, measurement angle = 173 degrees backscatter, equilibration time = 30 sec, cell = ZEN0040 (plastic disposable cuvettes), measurement duration = 10 sec, number of runs = 12, number of runs per measurement = 6, and run temperature = 25˚C. Size is reported as hydrodynamic radius (nm) by majority (99.6–99.9%) peak volume. Six individual measurements were performed for each data point (i.e., number of days in PBS at 37˚C). In addition, temperature ramping of both antibodies was performed using heat-DLS. Antibodies αF1Ig AM2 (0.74 mg/mL) and αF1Ig AM8 (1.49 mg/mL) were heated from 37˚C to 55˚C with 5-min equilibration times between temperature readings, with the other DLS conditions the same as described above.

**Y. *pestis* culture used for initial in vitro antibody characterization.** *Y. pestis* A1122 glycerol stocks, obtained from archived culture stocks in the Los Alamos National Laboratory, were streaked on tryptic soy broth agar with 5% sheep's blood (Teknova #T0148) and allowed to form distinct colonies by incubation at 26˚C overnight. Liquid cultures were prepared by inoculating single colonies in brain heart infusion (BHI) broth (BD Biosciences #221812) and overnight growth. F1-positive *Y. pestis* was cultured at 37˚C in BHI medium supplemented with 2.5 mM CaCl$_2$, whereas F1-negative *Y. pestis* cultures were grown at 23˚C without CaCl$_2$. Live cells were washed by multiple rounds of a) 10 min centrifugation at 8,500 g, b) removal of supernatant, and c) resuspension of cell pellet in PBS. *Y. pestis* cell fixation was achieved using cold 4% paraformaldehyde (Sigma-Aldrich #1004960700), followed by extensive PBS wash (at least 3 cycles). Absence of live cells (before using fixed cells in BSL1 conditions) was assessed by plating 100 μL fixed cell suspension on a TSB agar plate and checking for colony formation after overnight incubation at 26˚C. Fixed cells were stored at 4˚C for several months in the presence of 0.1% sodium azide.

**Kinetic characterization by whole-cell ELISA.** This assay used F1 expressed on the surface of *Y. pestis* grown at 37˚C and was performed as described before [41]. Briefly 1 x 10$^4$

fixed bacterial cells in 100 μL PBS per well were attached to a 96-well MaxiSorp ELISA plate (ThermoFisher #44240421) by overnight incubation at 4°C, followed by PBS washes and blocking with Wonder Block (WB; 0.3% BSA, 0.3% skimmed milk, and 0.3% fish gelatin in PBS). The primary antibody stocks (anti-F1 IgGs, or anti-influenza M2 IgG Z3 negative control [68] were serially diluted in 1:10 WB:PBS (light WB, LWB), and 100 μL dilution was added to each cell-coated and blocked well. After 1 h incubation, the antibody solution was removed, and the plate washed. HRP-conjugated anti-human antibody in LWB was added, and the plate incubated at 25°C for 1 hour. After washing, 100 μL/well TMB substrate (Sigma-Aldrich #CL07) was added to each well, the reaction was stopped by adding 100 μL/well 0.18 M $H_2SO_4$, and the absorbance at $\lambda$ 450 nm of each well was measured. Experiments were performed in triplicate, background absorbance (obtained with negative control antibody) was subtracted from each data point, and the averages of the resulting numbers (data points) plus standard deviations (error bars) were plotted against the corresponding primary-antibody concentrations using KaleidaGraph. Data were fitted to the one-site specific binding equation defined above (Eq #1). Comparative analysis between the original and affinity matured clones were conducted to assess improvement in affinity for the antigen.

**Kinetic characterization by SPR.** SPR analysis was performed by Carterra Inc. (Salt Lake City, UT) using their high-throughput instrument (LSA). Chip CMD200M (Carterra, #4287) was coated with protein A/G by priming the instrument with HBSTE running buffer (Hepes Buffer Saline with Tween and EDTA, Carterra, #3630), activating the chip surface for 8 min with 33 mM S-NHS and 133 mM EDC in 100 mM MES buffer pH 5.5, and injecting protein A/G (Pierce # 21186) for 10 min at 100 μg/mL in 10 mM NaAcOH pH 4.0. The chip surface was quenched with 1 M ethanolamine and conditioned by running 2 x 30 sec cycles of 10 nM glycine pH 2. Antibodies αF1Ig AM2 and 8 were added at eight different concentrations starting at 15 μg/mL plus seven 3-fold serial dilutions. Antibodies solutions were captured on the protein A/G chip in an array for 40 min. The F1 antigen (expressed in *E. coli*, BEI NR-44223 monomer enriched from BEI) was used at five different concentrations starting at 1 μM plus four 5-fold serial dilutions. After antibody capture a series of blank buffer injections were performed, followed by the various solutions of antigen. RBD from SARS-CoV-2 was used as a negative control (Acrobiosystems, #SPD-C52H3).

## In vitro and in vivo protection studies

**Bacterial strains and growth conditions used for in vitro macrophage-based assays and in vivo protection studies.** The *Y. pestis pgm-* pPst-, used for *in vitro* studies, was generated at USAMRIID and was kindly provided by Dr. Susan Welkos (USAMRIID, Fort Detrick, Frederick, MD) [35]. *Y. pestis pgm-* pPst-is an attenuated strain derived from the fully virulent *Y. pestis* CO92 strain, which is cured of the pPst plasmid containing the plasminogen activator (Pla) virulence locus (*pla*) and is pigmentation-deficient (*pgm*) [69–71]. This strain allows to perform moderate to high-throughput assays under BSL-2 condition for initial down selection. *Y. pestis pgm-* pPst- was grown on Remel® Sheep Blood Agar plates (Remel, Thermo Fisher #01202) and incubated at 37°C for 24 h. Bacterial colonies were harvested and used to inoculate 10 mL of brain heart infusion (BHI) broth (BD Biosciences #211059) and incubated in the BHI medium for 2 hours at 37°C with shaking at 200 rpm prior to infecting the macrophages. To ensure the bacteria were harvested in the log phase of growth the $OD_{600}$ of the culture post-incubation was not allowed to exceed 1.0 prior to incubation with antibodies.

**Mouse monoclonal antibodies.** The anti-F1 mouse monoclonal antibody F1-04-A-G1 was provided by provided by James Burans and Jennifer Adrich (Naval Medical Research Center, Silver Spring, MD) [35].

**Cell culture.** RAW264.7 murine macrophage-like cells derived from an Abelson murine leukemia virus tumor (ATCC TIB-71) were grown at 37˚C in 5% $CO_2$ in low glucose DMEM containing 10% fetal bovine serum, 1% L-glutamine, 1% non-essential amino acids, and 1% HEPES buffer. Cells were used before passage 15 and seeded in 96-well plates.

**Quantification of viable intracellular *Y. pestis* (Gentamicin Protection Assay).** Bacterial cultures were suspended in DMEM from cultures grown in brain heart infusion broth, and multiplicity of infection (MOI) was estimated for an $OD_{600}$ of 1.0 (~ 5.34 x $10^8$ colony forming units (CFU) per milliliter). As previously described for macrophage infection assays, RAW264.7 cells (1.5 x $10^4$ cell/well) were seeded into 96-well plates (Greiner Bio-OneT CellStarT µClearT 96-Well, Cell Culture-Treated, Flat-Bottom Microplate, # 655090) one day prior to infection [72]. *Y. pestis*, at 8 x $10^6$ CFU/mL, was pre-incubated with 10 µg/mL antibodies in DMEM for 1 hous at 37˚C prior to infection. Macrophages were then infected at an approximate MOI of 10 in triplicate wells and the plates were centrifuged at 200 x g for 5 min to initiate infection, and then placed into a 37˚C incubator with 5% $CO_2$. After 1 h of infection, gentamicin (8 µg/mL) was added to the wells to kill extracellular bacteria and the plates were incubated for an additional hour at 37˚C in the presence of 5% $CO_2$. After incubation, macrophages were washed two times in PBS and lysed using 0.1% Triton X-100 in PBS. Serial dilutions of lysates were plated in duplicate on sheep's blood agar and incubated for 2 days at 28˚C for CFU enumeration. The geometric means of CFU per mL were plotted for control samples not incubated with antibody compared to those incubated with 10 µg/mL of each antibody.

**Treatment of mice with antibodies and exposure of mice to aerosolized *Y. pestis*.** Female BALB/c mice what were 7–9 weeks old (Charles River, Frederick, MD) received antibodies via intraperitoneal injections approximately 18 hous prior to exposure to aerosolized *Y. pestis*. Mice treated with commercially available normal mouse IgG (Rockland antibodies and assays, Limerick, PA) were used as controls. Aerosolized challenge doses of virulent *Y. pestis* CO92 (pneumonic plague model) were prepared as follows. The bacteria were streaked from freezer stocks onto tryptose blood agar base slants (BD Biosciences, #223220) and grown for approximately 48 hours at 30˚C. Flasks of Heart infusion broth (HIB, BD Biosciences #23840) plus 0.2% xylose (Sigma Aldrich, St. Louis, MO, X1500) were inoculated with the bacterial growth from the freshly grown slants for an approximate initial $OD_{620}$ of 0.01 and were grown for approximately 24 hours at 30˚C. The cultures were harvested by centrifugation and suspended in HIB medium (no xylose) to the estimated concentration yielding the desired number of $LD_{50}$. Exposure of mice to aerosolized bacteria was accomplished as previously described [73,74]. Briefly, mice were transferred to wire mesh cages and up to four wire mesh cages were placed in a whole-body aerosol chamber within a class three biological safety cabinet located inside a BSL-3 laboratory. Mice were exposed to aerosolized *Y. pestis* strain CO92 created by a three-jet collision nebulizer. Samples were collected from the all-glass impinger (AGI) vessel and analyzed by performing CFU calculations to determine the inhaled dose of *Y. pestis*. The median lethal dose for *Y. pestis* CO92 in female BALB/c mice is approximately 6.8 x $10^4$ inhaled CFUs [74].

**Ethics statement.** Animal research at the United States Army Medical Research Institute of Infectious Diseases (USAMRIID) was conducted under an animal use protocol approved by the USAMRIID Institutional Animal Care and Use Committee (IACUC) in compliance with the Animal Welfare Act, PHS Policy, and other Federal statutes and regulations relating to animals and experiments involving animals. The facility where this research was conducted is accredited by the Association for Assessment and Accreditation of Laboratory Animal Care International (AAALAC) and adheres to principles stated in the Guide for the Care and Use of Laboratory Animals (National Research Council, 2011). Challenged mice were observed at least daily for 21 days for clinical signs of illness. Humane endpoints were used during all

studies, and mice were humanely euthanized when moribund, according to an endpoint criteria score sheet approved by the IACUC prior to beginning the study. Animals were scored on a scale of 0–16: 0–2 = no significant clinical signs (e.g., slightly ruffled fur); 3–7 = significant clinical signs such as subdued behavior, hunched appearance, absence of grooming, labored breathing of varying severity (increased monitoring was warranted); >8 = distress. Those animals receiving a score of > 8 were humanely euthanized by $CO_2$ exposure using compressed $CO_2$ gas or by injection of pentobarbital-based euthanasia solution; both methods of euthanasia were followed by cervical dislocation for confirmation of death. However, even with multiple observations per day, some animals died as a direct result of the infection.

**Statistics.** Gentamicin protection assay results were compared by stratified Wilcoxon (Van Elteren) test. The log-rank test was used to compare mouse survival curves post-challenge and Fisher's exact test was used to compare survival rates. Plaque counts were compared by stratified Wilcoxon (Van Elteren) test. Analysis was implemented in SAS version 9.4 (SAS Institute Inc., Cary, NC).

## Supporting information

**S1 Fig. Kinetic study of yeast-displayed antibody.** The yeast-associated fluorescence after incubation with biotinylated F1V and staining with streptavidin-APC (allophycocyanin) was measured by flow cytometry. Mean fluorescence intensity values at antigen concentrations of 500, 50, and 5 nM are shown for the original (αF1sc 2 and 8) and the progressively affinity matured antibody clones. The nomenclature EPX-Y denotes the error-prone library number (X) and the best clone number (Y) identified from this library. The best affinity matured αF1sc 2 clone (indicated for simplicity as αF1sc AM2) was clone 18 from error-prone library 3 (αF1sc 2 EP3-18). The best affinity matured αF1sc 8 clone (indicated as αF1sc AM8 for simplicity) was clone 24 from error-prone library 2 (αF1sc 8 EP2-24).
(PDF)

**S2 Fig. Kinetic study of purified anti-F1 IgGs by ELISA and SPR.** (**A**) Setup of whole cell enzyme-linked immunosorbent assay (ELISA, top) and surface plasmon resonance (SPR) kinetic study. (**B**) ELISA data obtained for original anti-F1 IgGs (αF1Ig 2 and 8) or affinity matured (αF1Ig AM2 and 8) antibodies are shown. Experiments were performed in triplicate and each data point averaged. The standard deviation of each data point was calculated, and shown as the error bar. Data were fitted to the one-site specific binding equation.
AB = antibody-F1 binding; $AB_{max}$ = antibody-F1 binding at saturation; $K_D$ = dissociation constant = half saturating antibody concentration ([Ab]). (**C**) SPR sensograms obtained for each parental and affinity matured antibody are shown. The dissociation constant and the errors shown in Table 1 were averages and standard deviations of the $K_D$ values measured in these SPR experiments.
(PDF)

**S1 File. Supplementary tables. Sequences of original and progressively affinity matured antibodies.** The amino acid sequence (one letter code) of single chain antibodies αF1sc 2 and 8 before affinity maturation and after various rounds of affinity maturations (AM) are shown.
(PDF)

**S1 Raw data. Data used to generate Fig 2.**
(XLSX)

**S2 Raw data. Data used to generate Fig 3.**
(XLSX)

**S3 Raw data. Data used to generate Fig 4.**
(XLSX)

**S4 Raw data. Data used to generate Fig 5.**
(XLSX)

## Acknowledgments

The antibody stability work was performed at the Center for Integrated Nanotechnologies, an Office of Science User Facility operated for the U.S. Department of Energy (DOE) Office of Science.

We thank BEI Resources, NIAID, NIH, for supplying the various forms of *Yersinia pestis* F1V fusion protein and Abel Ordonez Luna and Dr. Rekha Panchal for providing us with RAW264.7 cells. We also thank Madeline Bolding, Jenifer Zupancic and Lyman Monroe for the critical reading of the manuscript.

Opinions, interpretations, conclusions, and recommendations are those of the authors and are not necessarily endorsed by the U.S. Army or the Department of the Army Defense Health Agency.

## Author Contributions

**Conceptualization:** Nileena Velappan, Sergei S. Biryukov, Nathaniel O. Rill, Christopher P. Klimko, Raysa Rosario-Acevedo, Jennifer L. Shoe, Melissa Hunter, Jennifer L. Dankmeyer, David P. Fetterer, Daniel Bedinger, Mary E. Phipps, Rebecca J. Abergel, Armand Dichosa, Stosh A. Kozimor, Christopher K. Cote, Antonietta M. Lillo.

**Data curation:** Nileena Velappan, Sergei S. Biryukov, Nathaniel O. Rill, Daniel Bedinger, Christopher K. Cote, Antonietta M. Lillo.

**Formal analysis:** Nileena Velappan, Sergei S. Biryukov, Nathaniel O. Rill, Christopher P. Klimko, Raysa Rosario-Acevedo, Jennifer L. Shoe, Jennifer L. Dankmeyer, David P. Fetterer, Daniel Bedinger, Mary E. Phipps, Stosh A. Kozimor, Christopher K. Cote, Antonietta M. Lillo.

**Funding acquisition:** Stosh A. Kozimor, Christopher K. Cote.

**Investigation:** Nileena Velappan, Stosh A. Kozimor, Christopher K. Cote, Antonietta M. Lillo.

**Methodology:** Nileena Velappan, Sergei S. Biryukov, Nathaniel O. Rill, Christopher P. Klimko, Raysa Rosario-Acevedo, Jennifer L. Shoe, Melissa Hunter, Jennifer L. Dankmeyer, David P. Fetterer, Daniel Bedinger, Mary E. Phipps, Austin J. Watt, Rebecca J. Abergel, Armand Dichosa, Stosh A. Kozimor, Christopher K. Cote, Antonietta M. Lillo.

**Project administration:** Christopher K. Cote, Antonietta M. Lillo.

**Supervision:** Stosh A. Kozimor, Christopher K. Cote, Antonietta M. Lillo.

**Validation:** Nileena Velappan, Sergei S. Biryukov, Nathaniel O. Rill, Christopher P. Klimko, Raysa Rosario-Acevedo, Jennifer L. Shoe, Melissa Hunter, Jennifer L. Dankmeyer, David P. Fetterer, Daniel Bedinger, Mary E. Phipps, Austin J. Watt, Rebecca J. Abergel, Stosh A. Kozimor, Christopher K. Cote, Antonietta M. Lillo.

**Visualization:** Nileena Velappan, Christopher K. Cote, Antonietta M. Lillo.

**Writing – original draft:** Nileena Velappan, Christopher K. Cote, Antonietta M. Lillo.

**Writing – review & editing:** Nileena Velappan, Sergei S. Biryukov, Nathaniel O. Rill, Christopher P. Klimko, Raysa Rosario-Acevedo, Jennifer L. Shoe, Melissa Hunter, Jennifer L. Dankmeyer, David P. Fetterer, Daniel Bedinger, Mary E. Phipps, Austin J. Watt, Rebecca J. Abergel, Armand Dichosa, Stosh A. Kozimor, Christopher K. Cote, Antonietta M. Lillo.

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
