## [Decision Letter · Decision Letter 0]

4 Jan 2024

PONE-D-23-29661Characterization of Two Affinity Matured Anti-Yersinia pestis F1 Human Antibodies with Medical Countermeasure PotentialPLOS ONE

Dear Dr. Lillo,

Thank you for submitting your manuscript to PLOS ONE. After careful consideration, we feel that it has merit but does not fully meet PLOS ONE’s publication criteria as it currently stands. Therefore, we invite you to submit a revised version of the manuscript that addresses the points raised during the review process.

Please submit your revised manuscript by Feb 18 2024 11:59PM. If you will need more time to complete your revisions, please reply to this message or contact the journal office at plosone@plos.org. Please include the following items when submitting your revised manuscript:A rebuttal letter that responds to each point raised by the academic editor and reviewer(s). You should upload this letter as a separate file labeled 'Response to Reviewers'.A marked-up copy of your manuscript that highlights changes made to the original version. You should upload this as a separate file labeled 'Revised Manuscript with Track Changes'.An unmarked version of your revised paper without tracked changes. You should upload this as a separate file labeled 'Manuscript'.

We look forward to receiving your revised manuscript.

Kind regards,

Chandra Shekhar Bakshi, DVM, Ph.D.

Academic Editor

PLOS ONE

Journal Requirements:

"This research was supported by the Los Alamos National Laboratory (LANL) Directed Research

and Development Program (LANL LDRD 20180005DR), and the LANL Richard P. Feynman

Center for Innovation Strategic Investment Initiative. This work was performed, in part, at the

Center for Integrated Nanotechnologies, an Office of Science User Facility operated for the U.S.

Department of Energy (DOE) Office of Science. Los Alamos National Laboratory, an

affirmative action equal opportunity employer, is managed by Triad National Security, LLC for

the U.S. Department of Energy’s NNSA, under contract 89233218CNA000001. JPEO-CBRND

through chemical biological defense funding supported part of this work in an effort to analyze

prediscovered antibodies for efficacy against plague.

We thank BEI Resources, NIAID, NIH, for supplying the various forms of Yersinia pestis F1V

fusion protein and Abel Ordonez Luna and Dr. Rekha Panchal for providing us with RAW264.7 cells.

Opinions, interpretations, conclusions, and recommendations are those of the authors and are not

necessarily endorsed by the U.S. Army or the Department of the Army Defense Health Agency."

Funding information should not appear in the Acknowledgments section or other areas of your manuscript. We will only publish funding information present in the Funding Statement section of the online submission form. 

"AML - LANL LDRD 20180005DR Los Alamos National Laboratory (LANL) 

www.lanl.gov 

"AML - LANL LDRD 20180005DR Los Alamos National Laboratory (LANL) 

www.lanl.gov 

We note that one or more of the authors is affiliated with the funding organization, indicating the funder may have had some role in the design, data collection, analysis or preparation of your manuscript for publication; in other words, the funder played an indirect role through the participation of the co-authors. If the funding organization did not play a role in the study design, data collection and analysis, decision to publish, or preparation of the manuscript and only provided financial support in the form of authors' salaries and/or research materials, please do the following:

(1) Review your statements relating to the author contributions, and ensure you have specifically and accurately indicated the role(s) that these authors had in your study. These amendments should be made in the online form.

(2) Confirm in your cover letter that you agree with the following statement, and we will change the online submission form on your behalf: 

**Additional Editor Comments:**

The authors should adequately address the concerns expressed by the reviewers related to the explanation lacking the selection of F1 protein as a target for a therapeutic antibody as it is not a required virulence factor, bioweapons are designed to be F1 negative, and the areas where antibiotic-resistant strains are used do not have the resources to employ mAb therapy. None of these important points are discussed in the manuscript. Also, adequate explanations are needed to justify the system used and the deficiencies in the flow system used to select for binding as a function of the yeast cells that express the antibody.

Reviewers' comments:

Reviewer's Responses to Questions

**Comments to the Author**

1. Is the manuscript technically sound, and do the data support the conclusions?

Reviewer #1: Partly

Reviewer #2: Yes

2. Has the statistical analysis been performed appropriately and rigorously? 

Reviewer #1: I Don't Know

Reviewer #2: Yes

3. Have the authors made all data underlying the findings in their manuscript fully available?

Reviewer #1: No

Reviewer #2: Yes

4. Is the manuscript presented in an intelligible fashion and written in standard English?

Reviewer #1: No

Reviewer #2: Yes

5. Review Comments to the Author

Reviewer #1: The authors present a study on the further development of anti-F1 monoclonal antibodies for use as countermeasures against Yersinia pestis infection. The concept of using F1 as a target is questionable because, while it is a strong immunogen, it is not a required virulence factor for the pathogen, and the general consensus is that weaponized strains will be F1negative as the F1 protein is a principal component of western vaccine development. Additionally, the regions where naturally occurring antibiotic resistant strains are found are among the most resource limited areas in the world; areas where a very expensive technology, like a therapeutic monoclonal antibody could not be employed due to cost. These are items that really need to be discussed in the paper and are absent from it. Additional issues with the study are a general overestimation of the beneficial effects of their affinity maturation system, which is burdensome and had a negligible outcome on survival and affinity as determined by SPR, which is really the gold standard here. The writing in many areas is unclear, specific examples are given below. The way they analyze their binding data by flow cytometry doesn’t not appear to be a good reflection of the ability of the antibody to bind to the antigen, but rather a measure of binding as a function of expression level. If the cell could express the antibody at a high rate, then it bound a lot of antigen. However, that doesn’t mean that an antibody expressed at a lower rate doesn’t have a higher affinity. The exression level of the scFv in in yeast is irrelevant to antibody production because the CDRs are put into a different expression system later on for production of functional antibody. The metrics for defining what was considered good binding are not defined. It’s not really apparent that there were any. It seems that the investigators just drew a gate at the top of the dot plot cluster, but there doesn’t appear to be a defined cutoff.. Furthermore, the whole antibody production design is poorly explained in the paper. There are numerous systems that exist not where CDRs can be cloned into human IgGs that are optimized with glycosylation for Fc receptor interactions and increased uptake. The affinity of the binding domain is only one small part of the functional capacity of an antibody, and it really doesn’t seem like the whole process here improved on antibody binding anything more marginally. All of this reduces enthusiasm for the manuscript. More specific issues are given below.

The intro is wordy and can be shortened. There is a lot of superfluous information there. Mab therapy for COVID was relatively ineffectual unless given very early during infection, and efficacy dropped off through mutation of the spike protein. They did not play a significant role in the pandemic. Furthermore, going on about massive mutations of a functional antigen not being likely to happen doesn’t really have relevance to the current study, as F1 is not an antigen required for virulence, and F1 negative strains cause natural human disease and are the most likely kind of strain to be used in a bioterrorist attack to get around vaccines that have been developed.

Line 78, define F1V for the reader.

Line 118-120: What was considered a high signal, was there a specific MFI cutoff? Also, wouldn’t the PE signal be directly related to the Alexa 633 signal in any case where there is binding? MFI is a characteristic of the amount of fluorescence on a given particle, therefore by definition a high -633 signal would require a high PE signal. You shouldn’t have one without the other. In a perfect world you should have linearity between your 633 signal and your PE signal if the mutated antibody were binding to the F1 at saturation. Thus a clone that has higher expression would artificially be promoted as a clone with better binding, even if the actual affinity could be lower when those CDRs are put into an antibody. It’s also unclear to this reviewer why higher expression in a yeast system would be reflective of any characteristic for the antibody later on given that the variable regions were inserted into a cassette system later on? Perhaps expanded on this discussion a bit further.

Supplementary figure 1 should be in the paper with figure 2. Just showing the MFI in figure 2 alone doesn’t give great information because it is doesn’t account for differences in expression. The Y axis of figure 2 should be labelled MFI, not antigen binding.

The specific binding equation in the methods and materials needs some additional clarification. Throughout this the authors refer to antigen binding as a fluorescence value (633 fluorescence), which presumably is MFI though this is not clearly stated. Not sure if it is the value calculated using this equation. In the equation, AB max is maximum antigen/antibody binding, but what is that? Is that the highest MFI at the highest concentration of antigen? Is that just the MFI from a particular experiment? [ab] = antibody concentration, but concentration of what antibody? You don’t know the concentration of the yeast antibody so it must be the MFI of the SV tag antibody, which is not concentration. This is all unclear. Kd apparent dissociation constant using Kaleidagraph software. Better description of how this is calculated should be provided. It also needs to be defined, exactly where this equation was used because it isn’t apparent.

Line 159-161: ‘not the observed affinity improvements’ rather ‘the magnitude of the difference in the affinity measured’

Line 198-202: I would suggest that the difference is likely native vs. recombinant fusion protein. This idea should be expanded upon.

Line 203-211: SPR is the gold standard for affinity measurement. This should come before the ELISA data.

Line 216: by phagocytes, not just macrophages.

Line 222-223: move to discussion. In addition, have you considered the possibility that your antibodies induced a higher rate of intracellular killing of the Yersinia compared to the control. That would also read out as reduced bacteria following cell lysis.

237-239: You cannot extrapolate a human MaB therapy dose from mice, it’s not worth bothering doing so. They have different kinetics and metabolism of antibodies. Not only that, but you put a human antibody into a mouse, which will enhance it’s clearance. There are mab therapies that go into 150-200mgs in humans.

I’d much rather see survival curves than a table.

Your whole paper lacks even a mention of F1 negative pestis. It is very relevant here, and really needs to be included. F1 is a great antigen, it is fully protective at very low doses in a single shot in animal studies, but hasn’t gained approval because of the African Green Monkey studies at USAmRIID and the fact that it is completely ineffective against fully lethal F1 negative strains. Bioweapons will likely be F1 negative because it is a prominent vaccine target. The other issue, is the fact that mAb therapy is out of reach, cost-wise in the regions where antibiotic resistant bacteria are present. (Madagascar/central Africa). That will greatly limit usage. Its nice that you did the study showing that they are stable at 37C and 45C for long periods of time, but the overall impact of that finding is limited by the fact that they are serum proteins. IgG evolved through nature to exist at 37C for months.

Line 271: The technology is academically interesting, but the drawbacks haven’t been fully highlighted. It is labor intensive, and of the antibodies generated one showed no appreciable increase in affinity via SPR, the gold standard, and the one that did show an appreciable increase, worked worse in vivo. That’s not a great selling point.

Reviewer #2: This manuscript describes the maturation and characterization of mAbs as a potential therapeutic for pneumonic plague. As Y. pestis is categorized as a Category A agent, continuing research into mechanisms of pathogenicity as well as new treatment strategies is important to our defense.

Overall, the manuscript was clear. Please find my suggestions below, organized by Line:

87- replace ] with ) after technologies

172 – This sentence describes the direct output of your DLS measurements, specifically single peaks. However, that is not what is shown in Figure 3, which is referenced. Either consider adding your DLS scan output or move the reference to Figure 3 to the end of the sentence in Line 174.

179-180 – Why weren’t both IgGs subjected to the same DLS temperature scan? AM2 was analyzed at only 3 temperatures, whereas AM8 was analyzed at 7.

220 – There is a random e after the word either.

225 – Language should be modified for publication. Hours should also be spelled out not abbreviated h.

Conclusion – In general, the conclusion could be strengthened with specific result details and references that support why they are favorable findings. It is a bit bold to declare dose dependency of your mAb protection when your LD50s vary significantly across your experiments.

Figure 2 – Does this data represent combined experiments (if so how many) or is it representative (if so of how many)?

Figure 3 – Your methods state that the hydrodynamic radius of a control antibody was also measured. That should be included in figure 3 to demonstrate the variability in your IgGs over time is standard or better than standard.

Figure 5 – Statistical stars should be centered above data.

6. PLOS authors have the option to publish the peer review history of their article (what does this mean?). If published, this will include your full peer review and any attached files.

Reviewer #1: No

Reviewer #2: No

---

## [Author Response · Author response to Decision Letter 0]

27 Mar 2024

We thank the editor of PLOS1 and the reviewers for their critical reading of our manuscript. The document below contains comments from the reviewers in black, our response in blue and specific text changes to the manuscript in green.

Reviewer #1: The authors present a study on the further development of anti-F1 monoclonal antibodies for use as countermeasures against Yersinia pestis infection. The concept of using F1 as a target is questionable because, while it is a strong immunogen, it is not a required virulence factor for the pathogen, and the general consensus is that weaponized strains will be F1negative as the F1 protein is a principal component of western vaccine development. 

We thank the reviewer for the thorough review of this manuscript. We are well aware of the virulence associated with nonencapsulated strains of Y. pestis and our research continues to examine novel medical countermeasures to include both active and passive protection against F1-negative strains. However, in order to be prepared for all possible threats, we must be ready for all scenarios and we envision anti-F1 mAbs to be part of a combination or cocktail approach. To further emphasize the relevance of our antibodies, a patent has been granted recently (Synthetic anti-plague antibodies; Lillo AM, Velappan N, Dichosa AEL, Liley LM, Kozimor SA; USA 2023 Patent Number 11702465). This is the same rationale that resulted in a multi-antigen (or fusion protein) vaccine being developed in both the U.K. and U.S. Additionally, to date, natural infection with F1-negative strains is exceedingly rare, so an anti-F1 mAb would be quite applicable in a public-health context and with growing concerns for antimicrobial resistant strains. The reviewed version of the manuscript includes an additional paragraph and references to underscore the importance of the nonencapsulated strains (see below).

Additionally, the regions where naturally occurring antibiotic resistant strains are found are among the most resource limited areas in the world; areas where a very expensive technology, like a therapeutic monoclonal antibody could not be employed due to cost. These are items that really need to be discussed in the paper and are absent from it. 

The antibodies described in our manuscript have good resistance to aggregation at high temperatures for extended periods of time. Since refrigeration is one of the major issues facing the developing world in utilizing monoclonal antibody therapy, highly thermostable antibodies such as described here should increase utility of therapeutic monoclonal antibodies. Issues such as socioeconomic constraints in various parts of the world that could limit the development of medical technologies and medical countermeasures, are outside of the scope of this manuscript. We believe this concept to be subjective commentary and is more political or policy-driven in nature.

Additional issues with the study are a general overestimation of the beneficial effects of their affinity maturation system, which is burdensome and had a negligible outcome on survival and affinity as determined by SPR, which is really the gold standard here. The writing in many areas is unclear, specific examples are given below. […] The affinity of the binding domain is only one small part of the functional capacity of an antibody, and it really doesn’t seem like the whole process here improved on antibody binding anything more marginally.

In vitro affinity maturation, the counterpart of somatic hyper mutation in B cells, is an integral part of antibody engineering (https://www.ncbi.nlm.nih.gov/pmc/articles/PMC8726058/). The error-prone PCR based affinity maturation we employed in our work allows for amino acid changes in both CDR regions as well as the more conserved framework regions. These changes have the potential to impact antibody binding affinity and stability, often in unpredictable ways. Considering the great potential value of the maturation process we found it important to employ it for this work, and to describe it at an unprecedented level of detail to the display technology community. We consider this a very valuable component of our paper. 

In reference to the comment that the measured affinity improvements are minimal and that SPR is the gold standard assay that should count the most, please consider the following perspective. There are major differences in the antigen and antibody formats among the different assays performed for this work. The yeast display-based affinity measurements (and the maturation process) used F1V and scFv, SPR used F1 and IgG, and ELISA used Y. pestis-displayed F1V and IgG. We surmise that the difference in antigen formats, is the main culprit for the discrepancies in the affinity values and the observed low affinity increments (or lack-there-of). For example, it is entirely conceivable that during affinity maturation (where we used F1V as the target) we selected �F1sc 2 mutants with improved binding at the interphase of F1 and V antigens. The lack of V antigen in the monomeric F1 used in SPR might explain the higher KD values (lower affinity) measured using this technique. Additionally, although we agree that SPR is the gold standard in affinity measurements, the ELISA used here reflects a more real-world scenario (antibody interacting with Y. pestis-displayed F1 antigen) than the SPR assay. So it could be argued that �F1Ig AM2, which based on SPR did not have higher affinity than �F1Ig 2 has indeed been affinity matured (2-fold increase in affinity).

While we acknowledge that the affinity increment of our matured antibodies compared to the parental ones is not as high as we had hoped for, please notice that increased manufacturability is an additional goal of the maturation process. High level of display on the yeast surface is generally a proxy for high stability (e.g. thermostability), foldability and low aggregation (all associated to good manufacturability). Manufacturability, together with affinity, is a feature enriched in the maturation process. Therefore, matured antibodies are likely to have higher stability than the parental antibodies. Our data show that we have indeed achieved at least higher stability. Although the stability of matured antibody was not measured side by side with parental antibodies a previous publication of ours (https://pubmed.ncbi.nlm.nih.gov/33294421/) suggests that parental antibodies start to degrade after 2.5 weeks storage at 37�C, whereas the matured antibodies (this paper) show no sign of aggregation after 150 days. 

To reflect the points in this part of the rebuttal, we have amended the manuscript as follows: 

Lines 276-299 

AM2 vs �F1Ig 2. A major cause of this discrepancy could again be the difference in the antigen format used in the two assays. During flow-based kinetic studies (and the affinity maturation process) we used dimer-enriched F1V complex, whereas for SPR kinetics we used monomer-enriched F1. It is possible that �F1 antibody 2 binds at the intersection between F1 and V antigens and that the antibody structure changes during affinity maturation were tailored to the presence of V antigen. The lack of V protein in the antigen format used for the SPR assay (monomeric F1, not F1V) might explain why this assay failed to reveal affinity maturation. Also notice, that although SPR is the gold standard in affinity measurements, the ELISA used here reflects a more real-world scenario (i.e. antibody interacting with Y. pestis-displayed F1 antigen) than the SPR assay. Therefore, ELISA data should weigh more in the decision of whether �F1Ig AM2 has indeed been matured.

Based on these observations we could conclude that both matured IgGs have indeed higher affinity for F1 antigen than parental IgGs. We do recognize that the increment in affinity was not as high as expected, however please note that increased manufacturability is an additional goal of the maturation process. This is the reason why during maturation we sort yeast that are not only strong antigen binders but also strong displayers. High level of display on the yeast surface is generally a proxy for high stability (e.g. thermostability), foldability and low aggregation (all associated to manufacturability). Therefore, matured antibodies are likely to have higher stability than the parental antibodies. Although in this work the stability of matured antibodies was not measured side by side with the parental counterpart, we previously reported that �F1Ig 2 and 8 start to degrade after 2.5 weeks storage at 37�C, whereas matured �F1Ig AM2 and AM8 (this paper) show no sign of aggregation after 150 days.

In summary, we can safely conclude that our yeast display-based affinity maturation process was successful and resulted in improved variants of parental antibodies �F1Ig 2 and �F1Ig 8, i.e. �F1Ig AM2 and �F1Ig AM8.

The way they analyze their binding data by flow cytometry doesn’t not appear to be a good reflection of the ability of the antibody to bind to the antigen, but rather a measure of binding as a function of expression level. If the cell could express the antibody at a high rate, then it bound a lot of antigens. However, that doesn’t mean that an antibody expressed at a lower rate doesn’t have a higher affinity. 

Determination of antibody affinity by yeast display and flow cytometry analysis is a widely accepted method to determine antibody affinity. One essential requirement is that antibody concentration (determined by number of yeast cells and average number of antibody copies on their surface) is well below antigen concentration. In this way antigen concentration can be assumed to be constant during the interaction with yeast. Additionally, antibody-antigen binding is normalized by antibody display, so there is no bias due to display (see below). While differences are observed between the affinities of yeast-displayed scFvs measured by flow cytometry and the corresponding IgGs measured by SPR, we disagree that flow cytometry does not provide a helpful indication of the affinity of the reported scFvs. Due to the spacing on the surface of yeast, in this assay the affinity of scFvs is less affected by avidity than measurements from a bivalent IgG format. This results in a lower observed affinity than a format with higher avidity, but it allows analysis of the intrinsic affinity of the scFv, which improves during our affinity maturation process. The improvement in affinity of individual scFvs tested results from an increased affinity in this format, and clones (original and AM) have similar levels of expression on yeast (for example display level (PE mean) for original YP8, EP1-19 and EP2-8 is 685,694 and 767 respectively). Similar MFI for display means that the increase in MFI for antigen binding is due to increased affinity and not due to increase in display levels. This type of observation has been previously reported (see references below).

• Zupancic JM, Desai AA, Schardt JS, Pornnoppadol G, Makowski EK, Smith MD, Kennedy AA, Garcia de Mattos Barbosa M, Cascalho M, Lanigan TM, Tai AW, Tessier PM. Directed evolution of potent neutralizing nanobodies against SARS-CoV-2 using CDR-swapping mutagenesis. Cell Chem Biol. 2021 Sep 16;28(9):1379-1388.e7. doi: 10.1016/j.chembiol.2021.05.019. Epub 2021 Jun 24. PMID: 34171229; PMCID: PMC8223476

• Marcus WD, Lindsay SM, Sierks MR. Identification and repair of positive binding antibodies containing randomly generated amber codons from synthetic phage display libraries. Biotechnol Progr. 2006;22:919–22

• Simons JF, Lim YW, Carter KP, Wagner EK, Wayham N, Adler AS, Johnson DS. Affinity maturation of antibodies by combinatorial codon mutagenesis versus error-prone PCR. MAbs. 2020 Jan-Dec;12(1):1803646. doi: 10.1080/19420862.2020.1803646. PMID: 32744131; PMCID: PMC7531523.

The expression level of the scFv in in yeast is irrelevant to antibody production because the CDRs are put into a different expression system later on for production of functional antibody.[…] Furthermore, the whole antibody production design is poorly explained in the paper. There are numerous systems that exist not where CDRs can be cloned into human IgGs that are optimized with glycosylation for Fc receptor interactions and increased uptake.

In the ATUM process of scFv conversion to IgG the entire VL and VH (not just the CDRs) are inserted in an IgG scaffold, so the IgG productivity WILL be influenced by the mutation process.

The metrics for defining what was considered good binding are not defined. It’s not really apparent that there were any. It seems that the investigators just drew a gate at the top of the dot plot cluster, but there doesn’t appear to be a defined cutoff.

As stated above, during the maturation process we sort yeast with the highest level of antigen binding and surface display. When the yeast population has an oblong shape, the right tip of the population contains the yeast of interest.

 [….] All of this reduces enthusiasm for the manuscript. More specific issues are given below.

Due to all the above-mentioned factors, we strongly believe that our decision to undertake affinity maturation of the anti-F1 parental antibodies is highly justified and very relevant to research focusing developing antibody-based medical countermeasures. 

The intro is wordy and can be shortened. There is a lot of superfluous information there. Mab therapy for COVID was relatively ineffectual unless given very early during infection, and efficacy dropped off through mutation of the spike protein. They did not play a significant role in the pandemic.

We feel that every element of the introduction ties to concepts described later in the paper. This is true of the discussion of the Covid antibodies as well. This particular example illustrates the concept of antibody cocktails (and need-there-of) and puts the use of antibody therapeutics in the context of a real work scenario. Of course, these are discussion points and not an endorsement of any particular product.

Furthermore, going on about massive mutations of a functional antigen not being likely to happen doesn’t really have relevance to the current study, as F1 is not an antigen required for virulence, and F1 negative strains cause natural human disease and are the most likely kind of strain to be used in a bioterrorist attack to get around vaccines that have been developed.

We are well aware of the concerns about nonencapsulated strains and have devoted much of our research to this concept, but we disagree with the reviewer on several points. F1-negative strains very rarely cause human infections. While F1- strains have been isolated in nature we are only aware of a single documented case of these strains causing human disease. Even this case (Winter, 1960) could have been the result of an infection with a “weakly” encapsulated strain as opposed to a truly nonencapsulated strain. Please see below for the information added to the manuscript regarding nonencapsulated strains.

Line 78, define F1V for the reader.

Thank you for the suggestion, we have edited the document to add F1V (F1 and V antigen of Y. pestis)

Line 118-120: What was considered a high signal, was there a specific MFI cutoff? Also, wouldn’t the PE signal be directly related to the Alexa 633 signal in any case where there is binding? MFI is a characteristic of the amount of fluorescence on a given particle, therefore by definition a high -633 signal would require a high PE signal. You shouldn’t have one without the other. In a perfect world you should have linearity between your 633 signal and your PE signal if the mutated antibody were binding to the F1 at saturation. Thus a clone that has higher expression would artificially be promoted as a clone with better binding, even if the actual affinity could be lower when those CDRs are put into an antibody. It’s also unclear to this reviewer why higher expression in a yeast system would be reflective of any characteristic for the antibody later on given that the variable regions were inserted into a cassette system later on? Perhaps expanded on this discussion a bit further.

Our experience and observations with yeast displayed scFvs suggest that scFvs with higher levels of display are more stable. However, the antigen bind

---

## [Editor Report · Decision Letter 1]

23 May 2024

Characterization of Two Affinity Matured Anti-Yersinia pestis F1 Human Antibodies with Medical Countermeasure Potential

PONE-D-23-29661R1

Dear Dr. Lillo,

We’re pleased to inform you that your manuscript has been judged scientifically suitable for publication and will be formally accepted for publication once it meets all outstanding technical requirements.

Kind regards,

Chandra Shekhar Bakshi, DVM, Ph.D.

Academic Editor

PLOS ONE

Additional Editor Comments (optional):

The authors have comprehensively addressed all the concerns from the previous review.
---

## [Editor Report · Acceptance letter]

24 Jun 2024

PONE-D-23-29661R1 

PLOS ONE

Dear Dr. Lillo, 

I'm pleased to inform you that your manuscript has been deemed suitable for publication in PLOS ONE. Congratulations! Your manuscript is now being handed over to our production team.

Kind regards, 

on behalf of

Dr Chandra Shekhar Bakshi 

Academic Editor

PLOS ONE